# Retinoic acid signaling regulates spatiotemporal specification of human green and red cones

Sarah E. Hadyniak[1], Joanna F. D. Hagen[1]☯, Kiara C. Eldred[1,2]☯, Boris Brenerman[1], Katarzyna A. Hussey[1], Rajiv C. McCoy[1], Michael E. G. Sauria[1], James A. Kuchenbecker[3], Thomas Reh[2], Ian Glass[2], Maureen Neitz[3], Jay Neitz[3], James Taylor[1†], Robert J. Johnston, Jr[1]*

1 Department of Biology, Johns Hopkins University, Baltimore, Maryland, United States, 2 Department of Biological Structure, University of Washington, Seattle, Washington State, United States, 3 Department of Ophthalmology, University of Washington, Seattle, Washington State, United States

☯ These authors contributed equally to this work.
† Deceased.
* robertjohnston@jhu.edu

**Data Availability Statement:** Bulk organoid, WERI-Rb-1 RNA sequencing, and association study sequencing and genotype data are available upon request from Johns Hopkins University Data

## Abstract

Trichromacy is unique to primates among placental mammals, enabled by blue (short/S), green (medium/M), and red (long/L) cones. In humans, great apes, and Old World monkeys, cones make a poorly understood choice between M and L cone subtype fates. To determine mechanisms specifying M and L cones, we developed an approach to visualize expression of the highly similar *M-* and *L-opsin* mRNAs. *M-opsin* was observed before *L-opsin* expression during early human eye development, suggesting that M cones are generated before L cones. In adult human tissue, the early-developing central retina contained a mix of M and L cones compared to the late-developing peripheral region, which contained a high proportion of L cones. Retinoic acid (RA)-synthesizing enzymes are highly expressed early in retinal development. High RA signaling early was sufficient to promote M cone fate and suppress L cone fate in retinal organoids. Across a human population sample, natural variation in the ratios of M and L cone subtypes was associated with a noncoding polymorphism in the *NR2F2* gene, a mediator of RA signaling. Our data suggest that RA promotes M cone fate early in development to generate the pattern of M and L cones across the human retina.

## Introduction

Trichromacy in humans is enabled by three subtypes of cone photoreceptors that express opsin photopigments sensitive to short (S), medium (M), or long (L) wavelengths of light. Among placental mammals, only humans, great apes, Old World primates, and some New World monkeys possess M and L cones [1–3]. The only known difference between M and L cones is the expression of their opsin photopigment, OPN1MW (M-opsin), or OPN1LW (L-opsin) [4]. M and L cones have been technically challenging to distinguish due to the high

Services (dataservices@jhu.edu). These data have not been deposited to a publicly available repository because patients involved in the association study did not consent to include the release of raw sequencing or genotype data. The pipeline to analyze the M- versus L-opsin pileups is available at https://github.com/bbrener1/johnston_retina. The pipeline to analyze the association data is available at https://github.com/bxlab/2018.09.01_Eldred_GWAS/releases/tag/v1.0. All other relevant data are within the paper and its Supporting Information files. All reagents are available upon request to the authors which can be directed to Robert Johnston Jr. (robertjohnston@jhu.edu). There are restrictions to the availability of cell lines listed in Table 1 due to Material Transfer Agreement (MTA).

**Funding:** S.E.H. was an F31 Ruth L. Kirschstein NRSA Fellow and supported by the National Eye Institute under grant 5F31EY029157-02. J.F.D.H was supported by the EMBO Postdoctoral Fellowship ALTF 318-2021. K.C.E. was supported by the HHMI Gilliam Fellowship, the National Science Foundation Graduate Research Fellowship Program under grant 174689, the Damon Runyon Cancer Research Foundation (DRG-# 32-20), and the Hanna H. Gray Fellows Program Award from the HHMI (GT15994). K.A.H. was supported by the National Science Foundation Graduate Research Fellowship Program under grant 1746891. R.C.M. was supported by R35GM133747 from NIH/NIGMS. M.N. and J.N. were supported by NEI grants R01EY021242 and P30EY001730, and unrestricted funds from Research to Prevent Blindness to ophthalmology at UW. R.J.J. was supported by the National Eye Institute R01EY030872, the BrightFocus foundation G2019300, and MSCRF 140841.

**Competing interests:** The authors have declared that no competing interests exist.

**Abbreviations:** ALDH, aldehyde dehydrogenase; ATRA, all-trans retinoic acid; CPM, counts per million; ERG, electroretinogram; FP-ERG, flicker-photometric electroretinogram; LCR, locus control region; LD, linkage disequilibrium; NDRI, National Disease Research Interchange; NRL, neural retina leucine; ONL, outer nuclear layer; RA, retinoic acid; SEM, standard error of the mean; THRB, thyroid hormone receptor beta; TPM, transcripts per million.

sequence and structural similarity between the M- and L-opsin mRNA transcripts and proteins, and thus, the mechanisms underlying this fate decision have proven elusive.

In humans, the *OPN1LW* (*L-opsin*) and *OPN1MW* (*M-opsin*) genes lie in a tandem array on the X chromosome under the control of a shared regulatory DNA element called a locus control region (LCR) [5–8]. Though the number and ordering of *OPN1LW* and *OPN1MW* genes varies among individuals, the LCR is most commonly found upstream of *OPN1LW*, followed by two copies of *OPN1MW* [9]. Regardless of the arrangement, only the first two opsin genes in the array are expressed [4,5].

Two nonexclusive models for M and L cone fate specification have been proposed. The stochastic model, which suggests that cones randomly choose M or L cone fate, is based on transgene reporter experiments in mice [6,10]. The temporal model, supported by functional microspectrophotometry, multifocal electroretinograms (ERGs), and qPCR-based expression studies, suggests that the late-born retinal periphery is enriched for L cones compared to the early-born central retina [11–13]. In this study, we examined human retinas and manipulated human retinal organoids to assess these mechanisms.

Organoids are powerful systems to investigate the molecular mechanisms controlling cell fate specification during human development [14]. We previously showed that thyroid hormone regulates the S versus M/L cone fate decision in human retinal organoids [15]. Here, we studied the development of human retinas and organoids and conducted an association study to understand the M versus L cone fate decision. Our data suggest that retinoic acid (RA) signaling controls the spatiotemporal patterning of M and L cones in the developing human retina.

## Results

### Visualization of *M-* and *L-opsin* expression

To distinguish M versus L cone fates, we developed a method to visualize *M-* and *L-opsin* mRNA expression. The opsin protein sequences are approximately 96% identical and the mRNA sequences are approximately 98% identical, complicating expression analysis. To directly visualize *M-* and *L-opsin* expression, we generated a 40-nucleotide probe for *M-opsin* and a 42-nucleotide probe for *L-opsin* with partial overlap that target a region with 8 differential nucleotides (**Fig 1A and 1B**). We utilized colorimetric in situ hybridization to visualize *M-opsin* in blue and *L-opsin* in pink.

We tested the specificity of this approach to distinguish *M-* and *L-opsin* mRNA by transfecting plasmids driving *M-opsin* and/or *L-opsin* and visualizing expression in HEK293 cells. We scored the total number of transfected cells that expressed *M-opsin* mRNA (**Fig 1C**, "M"), *L-opsin* mRNA (**Fig 1C**, "L"), or both *M-opsin* and *L-opsin* mRNA (**Fig 1C**, "M+L") for each experiment. HEK293 cells transfected with no plasmid showed almost no signal (**Fig 1C and 1D**). Cells transfected with a plasmid driving *M-opsin* showed expression of *M-opsin* but not *L-opsin* (**Fig 1C and 1E**), while cells transfected with a plasmid driving *L-opsin* displayed expression of *L-opsin* but not *M-opsin* (**Fig 1C and 1F**). Cells transfected with either a plasmid driving *M-opsin* or a plasmid driving *L-opsin* and then mixed, showed expression of *M-opsin* only or *L-opsin* only in individual cells (**Fig 1C and 1G**). Finally, cells co-transfected with both a plasmid driving *M-opsin* and a plasmid driving *L-opsin* displayed co-expression of *M-* and *L-opsin* (**Fig 1C and 1H**). In this experiment, cells expressing *M-opsin* or *L-opsin* only were also observed, due to variation in transfection of the two plasmids (**Fig 1C**). Thus, this in situ hybridization method distinguishes between the highly similar *M-* and *L-opsin* mRNAs.

We next related expression of opsin mRNA and protein. Due to the high sequence similarity of the M-opsin and L-opsin proteins, available antibodies detect both proteins. We

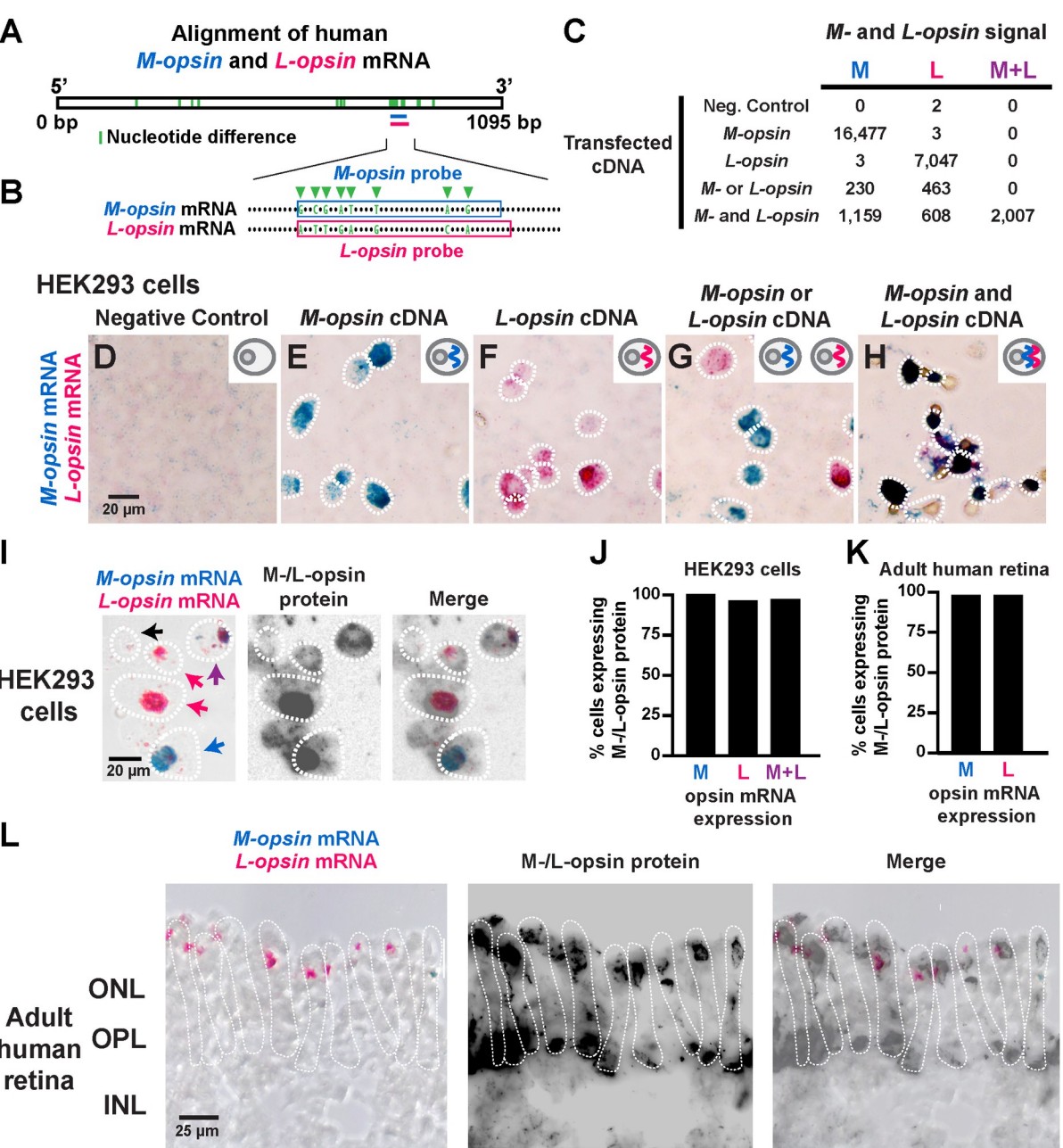

**Fig 1. An in situ hybridization approach distinguishes *M-opsin* and *L-opsin* mRNA. (A)** Alignment of human *M-opsin* and *L-opsin* mRNA. Green bar = nucleotide difference. Horizontal pink and blue lines = location of in situ hybridization probes. **(B)** Alignment of portions of exon 5 from *M-* and *L-opsin*. In situ hybridization probes target mRNA sequences, indicated by blue (*M-opsin*) and pink (*L-opsin*) boxes. Green arrowheads indicate 8 nucleotide differences. Dots indicate nucleotide alignment between the opsins. **(C–H)** HEK293 cells probed for *M-opsin* mRNA (blue) and *L-opsin* mRNA (pink). Insets = schematic of transfected plasmid. Cells that did not express *M-opsin* mRNA or *L-opsin* mRNA were not quantified. **(C)** Quantification of transfected HEK293 cells expressing *M-opsin* mRNA only, *L-opsin* mRNA only, or *M-opsin* mRNA and *L-opsin* mRNA for the conditions in **(D–H)**. **(D–H)** Brightfield images of cells with: **(D)** No plasmid transfected. **(E)** Transfection of a plasmid driving *M-opsin*. **(F)** Transfection of a plasmid driving *L-opsin*. **(G)** Transfection of either a plasmid driving *M-opsin* or a plasmid driving *L-opsin* independently and then the cells were mixed. **(H)** Transfection of both a plasmid driving *M-opsin* and a plasmid driving *L-opsin*. **(I)** Visualization of *M-opsin* mRNA, *L-opsin* mRNA, and M-/L-opsin protein (black) in HEK293 cells transfected with both a plasmid driving *M-opsin* and a plasmid driving *L-opsin*. *M-opsin* (blue) and *L-opsin* (pink). Blue arrow indicates a cell expressing *M-opsin* mRNA only. Pink arrows indicate cells expressing *L-opsin* mRNA only. Purple arrow indicates a cell expressing both *M-opsin* mRNA and *L-opsin* mRNA. Black arrow indicates an untransfected cell. Cells were identified based on nuclear Hoechst staining (S1A Fig). With this combined RNA in situ hybridization/immunohistochemistry approach, the mRNA signal was reduced, when compared to the mRNA signal observed when RNA in situ hybridization was conducted alone (Fig 1H). **(J)** Quantification of *M-/L-opsin* mRNA and M/L-opsin protein expression in transfected HEK293 cells **(I)**. Original data sets are in S1 Data. **(K)** Quantification of

*M-/L-opsin* mRNA and M/L-opsin protein expression in adult human retina (**L**). Original data sets are in S1 Data. (**L**) Visualization of *M-opsin* mRNA, *L-opsin* mRNA, and *M-/L-opsin* protein in cone cells in an adult human retina. *M-opsin* (blue) and *L-opsin* (pink). No cones co-expressed *M-opsin* mRNA and *L-opsin* mRNA. ONL, outer nuclear layer; OPL, outer plexiform layer; INL, inner nuclear layer. Cell boundaries were determined by identifying layers from a nuclear Hoechst stain (**S1B Fig**) and analyzing opsin protein immunohistochemistry signal from the ONL to the OPL.

transfected HEK293 cells with both a plasmid driving *M-opsin* and a plasmid driving *L-opsin* and observed cells that expressed *M-opsin* only, *L-opsin* only, or both *M-opsin* and *L-opsin* (**Fig 1I**), like our previous experiment (**Fig 1C and 1H**). We examined protein expression using immunohistochemistry with an M-/L-opsin antibody and observed expression of M-/L-opsin protein in nearly all cells that expressed *M-opsin* alone, *L-opsin* alone, or both *M-opsin* and *L-opsin* (**Figs 1I, 1J, S1A, and S1C**), showing that opsin mRNA-expressing cells identified by this method also express opsin protein.

We next tested this assay in adult human retinas. We identified cells that exclusively expressed *M-opsin* or *L-opsin* (**Fig 1K**). M-/L-opsin protein was observed in nearly all cells that expressed *M-opsin* or *L-opsin* mRNA (**Figs 1K, 1L, S1B, and S1C**), suggesting that this method identifies opsin-expressing M and L cones.

Together, we conclude that this in situ hybridization method distinguishes between *M-* and *L-opsin* mRNAs and reliably identifies M cones and L cones in adult human retinas.

## *M-opsin* is expressed before *L-opsin* during development

To assess the timing of M and L cone specification during development, we evaluated *M-opsin* or *L-opsin* mRNA in a 122-day-old human fetal retina and a 130-day-old human fetal retina. We predicted two main, alternate outcomes: (1) a mix of cones expressing either *M-opsin* or *L-opsin* if the decision was stochastic; or (2) a majority of cones expressing *M-opsin* or *L-opsin* only if the decision was temporal and one cone subtype was generated first. In the 122-day-old retina, we observed *M-opsin* and no *L-opsin* (100% *M-opsin* only cells, *n* > 500 cells) (**Fig 2A and 2B**). This retina displayed M-/L-opsin protein in the ONL (**S2A Fig**), suggesting that these cells were terminally differentiating M cones. This retina also displayed Ki67 expression (**S2A Fig**), suggesting that there are proliferating cells, ongoing development, and generation of additional neurons at this time point in this region of the fetal retina. In the 130-day-old retina, we observed >99% *M-opsin* only cells with sparse *L-opsin* only cells (**Fig 2A and 2B**). Opsin expression occurred within the outer nuclear layer (ONL, **S2I Fig**) and spanned an approximately 3 mm wide central region of the retina (approximately 38% of the total width of the globe), skewed towards the temporal side, between 1.6 and 4.5 mm from the most temporal edge (**S2C–S2H Fig**). The highest number of *M-opsin* expressing cells were detected in the center of this range, approximately 3 mm from the temporal edge (**S2H Fig**), consistent with observations indicating that the human retina develops from the center to the periphery [2]. Together, these data suggested that M cones are generated before L cones during human retinal development.

We next quantified *M-* and *L-opsin* expression in RNA-seq data sets by comparing counts of aligned sequencing reads for *M-* or *L-opsin* based on 20 nucleotide differences that distinguish these genes (**Fig 1A**), normalizing by the total number of aligned reads per sample ("normalized pileup count"). For these experiments, each point represents the normalized value for an individual nucleotide difference between *M-opsin* mRNA or *L-opsin* mRNA in an individual sample. We validated this approach in the WERI-Rb-1 retinoblastoma cell line, which expresses both *M-opsin* and *L-opsin* upon addition of thyroid hormone (T3) (**S2B Fig**).

We evaluated expression in 12 prenatal samples from day 52/54 through day 136 postconception from published RNA-seq data sets [16]. *M-opsin* expression was observed from day

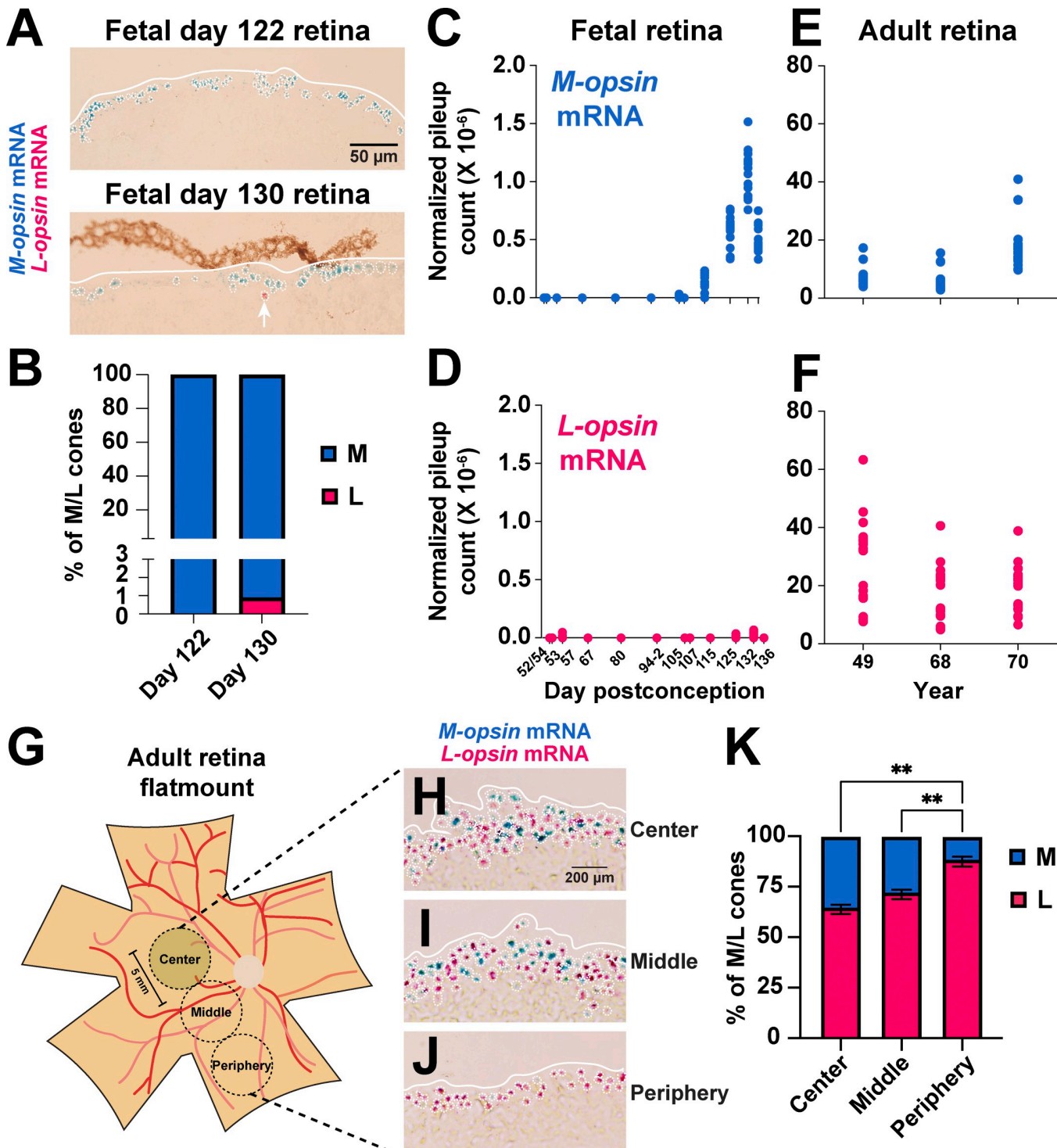

**Fig 2. Expression of *M-opsin* mRNA and *L-opsin* mRNA in fetal and adult human retinas.** (A) 20 μm sections of 122-day-old and 130-day-old human fetal retinas. *M-opsin* (blue) and *L-opsin* (pink). White arrow indicates *L-opsin*-expressing cell. (B) Quantification of % M and L cones in 122-day-old and 130-day-old human fetal retinas. (C–F) *M*- and *L-opsin* mRNA expression in developing human fetal retinas and adult retinas. Values indicate total pileup count, normalized to total read count. Each data point indicates detection of *M*- or *L-opsin* mRNAs, based an individual nucleotide difference. N = 1 for each time point, except 94–2 where N = 2. 52/54 = exact date is unclear. Analyzed from [16,17]. Original data sets are in S2 Data. (C) *M-opsin* mRNA in fetal retinas. (D) *L-opsin* mRNA in fetal retinas. (E) *M-opsin* mRNA in adult retinas. (F) *L-opsin* mRNA in adult retinas. (G) Schematic of adult human retina with regions isolated using a 5 mm biopsy punch. White circle = optic nerve. Red lines = blood vessels. Yellow circle = macular pigment. (H–J) 20 μm sections were probed for *M-opsin* (blue) and *L-opsin* (pink) mRNA. (K) Average ratios of M and L cones as percent of M/L total cones across 3 individuals. One-way ANOVA with

Tukey's multiple comparisons test: Center L versus Middle L = no significance; Center L versus Periphery L $p < 0.01$; Middle L versus Periphery L $p < 0.01$. ** Indicates $p < 0.01$. Original data sets are in S2 Data.

115 through day 136 (**Fig 2C**). *M-opsin* expression increased in retinas from day 115 through 132. *M-opsin* expression was present but lower on day 136 (**Fig 2C**). As these expression analyses were conducted on single human fetal retinal samples [16], the decrease is likely due to biological variability between samples. In contrast, *L-opsin* expression was minimally detected during this time (**Fig 2D**). We analyzed opsin expression from three independent adult human retinas from published data sets [17] and observed expression of both *M-* and *L-opsin* (**Fig 2E and 2F**). These data suggested that *M-opsin* is expressed before *L-opsin* during human development.

To assess cone distributions in the adult human retina, we dissected three retinas, used a biopsy punch to isolate 5 mm wide regions of tissue from the center, middle, and peripheral retina (**Figs 2G and S3A**), sectioned the tissue, and performed our in situ hybridization strategy. For each retina, we analyzed one punch from the center region, three middle regions, and three peripheral regions (**S3A Fig**). We manually quantified cells expressing *M-opsin* mRNA or *L-opsin* mRNA and validated our approach using semi-automated image analysis software (**S3B–S3E Fig**). The center and middle had significantly higher proportions of M cones compared to the periphery (**Figs 2H–2K and S3F–S3H**). As cone specification occurs from the center to the periphery during development [18,19], the higher proportions of M cones in the earlier-specified center and middle regions are consistent with the generation of M cones before L cones.

## Early retinoic acid signaling promotes M cones and suppresses L cones in human retinal organoids

We next sought to identify a mechanism that regulates M and L cone specification. RA signaling controls expression of an independently evolved opsin gene array in zebrafish [20–22]. We hypothesized that RA may play a role in the regulation of the M/L-opsin gene array in the developing human retina. We assessed expression of RA pathway regulatory genes in 12 prenatal samples from day 52/54 through day 136 postconception from published RNA-seq data sets [16]. Specific aldehyde dehydrogenases (ALDHs) convert retinaldehyde to RA. *ALDH1A1* and *ALDH1A3* are expressed early in retinal development and decrease over time, whereas *ALDH1A2* is expressed at very low levels throughout gestation (**Fig 3A**). Similar to *ALDH1A1* and *ALDH1A3* in humans (**Fig 3A**), *ALDH* orthologs in mouse [23], zebrafish [24], and chicken [25], are expressed highly early and decrease during development. Cytochrome p450 family 26 (CYP26) enzymes catalyze the degradation of RA. *CYP26A1* is steadily expressed at low levels, whereas *CYP26B1* and *CYP26C1* are not expressed (**S4A Fig**). The high expression of 2 RA-synthesizing enzymes early suggested that RA signaling is high early and decreases during development.

We next assessed spatiotemporal expression of RA pathway regulatory genes in samples taken from the center or periphery through fetal development from published RNA-seq data sets [16]. *ALDH1A1* and *ALDH1A3* were expressed highest in the center on day 59 and decreased on days 96 and 132 (**Fig 3B**). *ALDH1A2* was not expressed in the center (**Fig 3B**). In the periphery, *ALDH1A1* and *ALDH1A3* were expressed highest on day 59 (**Fig 3B and 3C**), with decreases on days 73, 96, and 132 (**Fig 3C**). *ALDH1A2* was not expressed in the periphery (**Fig 3C**). *CYP26A1* was steadily expressed in the center and periphery and *CYP26B1* and *CYP26C1* were not expressed (**S4B and S4C Fig**). Together, these data are consistent with (1) the overall decreases in *ALDH1A1* and *ALDH1A3* expression during human fetal retinal development (**Fig 3A**); and (2) high RA signaling early and decreasing during development.

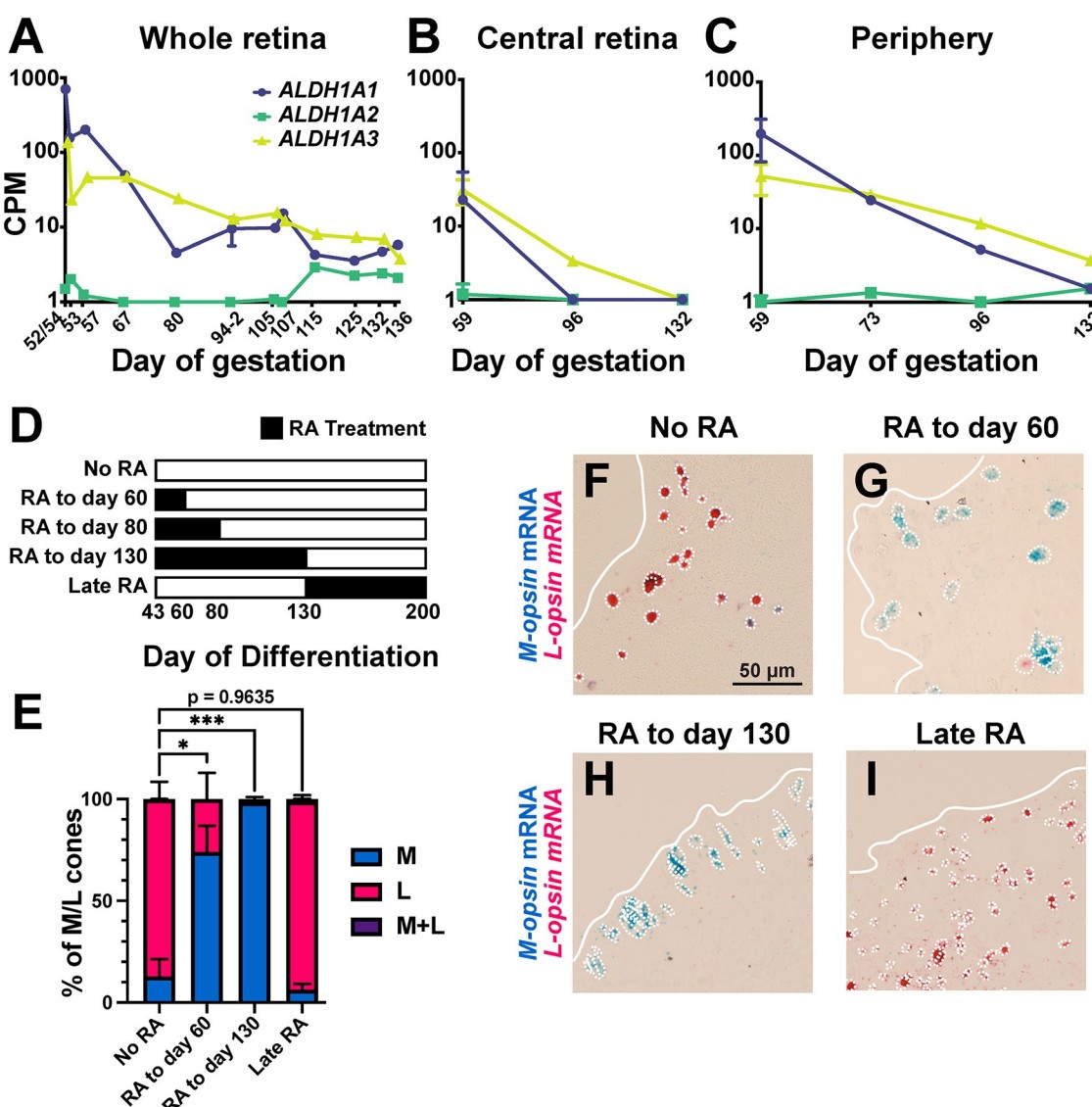

**Fig 3. RA signaling induces *M-opsin* and inhibits *L-opsin* early in human retinal organoids.** (**A–C**) Expression of *ALDH1A1*, *ALDH1A2*, and *ALDH1A3* in fetal human retinas by day of gestation and retinal region. CPM, log counts per million. Analyzed from [16]. Error bars for the 2 samples from fetal day 94 indicate SEM. Original data sets are in S3 Data. (**A**) Whole retina. (**B**) Central retina. (**C**) Periphery. (**D**) Black bars indicate temporal windows of 1.0 μm RA addition during retinal organoid culture. (**E**) Quantification of M and L cone ratios for RA treatments. For "No RA," $N = 3$; for "RA to day 60," $N = 6$; for "RA to day 130," $N = 3$; and for "Late RA," $N = 5$. One-way ANOVA with Dunnett's multiple comparisons test: "No RA" *L-opsin* versus "RA to day 60" *L-opsin*, $p < 0.05$; "No RA" *L-opsin* versus "RA to day 130" *L-opsin* $p < 0.005$; "No RA" *L-opsin* versus "Late RA" *L-opsin* $p = 0.9635$. Error bars indicate SEM. * Indicates $p < 0.05$; *** indicates $p < 0.005$. (**F–I**) *M-opsin* (blue) and *L-opsin* (pink) expression in organoids grown in different RA conditions (**D**), quantified in (**E**). White dotted outlines indicate *M-* or *L-opsin*-expressing cells. White lines indicate the edge of the organoid. (**F**) No RA. (**G**) RA to day 60. (**H**) RA to day 130. (**I**) Late RA.

As *M-opsin* is expressed before *L-opsin* and the temporal dynamics of *ALDH1A1* and *ALDH1A3* suggest high RA signaling early, we hypothesized that high RA signaling early promotes M cone fate and suppresses L cone fate. We tested this hypothesis using human retinal organoids. Differentiation of human retinal organoids involves addition of 1.0 μm RA on days 20 to 43 to promote early retinal patterning (S4D Fig). Organoids were grown in media without exogenous RA from day 43 to day 200, from the end of primitive retina differentiation through early cell fate specification (13% M, 87% L, 0.1% co-expressing M+L; Fig 3D–3F, "No

**RA"**). These organoids displayed a mix of M cones and L cones (**Fig 3E and 3F**), similar to adult human retinas (**Fig 2H–2K**).

To test whether RA was sufficient to promote M cone specification in retinal organoids, we added 1.0 μm RA over different timeframes and assessed M and L cone ratios at day 200 (**Fig 3D**). We previously examined gene expression at multiple time points during organoid development using bulk RNA-seq [15]. We analyzed this data and found that expression of thyroid hormone receptor beta (*THRB*), a marker of cone fate, stabilizes on day 70 (**S4E Fig**), suggesting that the generation of new cones ends around this time point. In contrast, expression of neural retina leucine zipper (*NRL*), a marker of later-born rod photoreceptors, stabilizes around day 160 (**S4E Fig**). Opsin expression was first observed at day 130 [15], suggesting that terminal differentiation of cones begins at this time point. We defined three main timeframes of cone development: immature cone generation through day 70, maturation from days 70 to 130, and terminal differentiation from day 130 onwards.

Addition of RA early through cone maturation from days 43 to 130 yielded organoids with almost exclusively M cones at day 200 (98% M, 2% L, 0% co-expressing M+L; **Fig 3E and 3H**; **"RA to day 130"**), suggesting that addition of RA through the beginning of *M*- and *L-opsin* expression was sufficient to promote M cone fate. To determine whether RA could promote M cone fate in immature cones, we added RA through day 60, before the end of immature cone generation and well before terminal differentiation and *M*- and *L-opsin* expression on day 130. In this condition, organoids were enriched for M cones (74% M, 26% L, 0% co-expressing M +L; **Fig 3E and 3G**; "RA to day 60"), suggesting that RA was sufficient to promote M cone fate in immature cones. Addition of RA late in development during terminal differentiation from days 130 to 200 yielded L cone-enriched organoids at day 200 (6% M, 93% L, 1% co-expressing; **Fig 3E and 3I**; **"Late RA"**), similar to "No RA" control organoids (**Fig 3E and 3F**), suggesting that RA was not sufficient to convert L cones to M cone fate.

For most experimental conditions, we observed minimal differences between densities of M and L cones (**S4F Fig**), suggesting that the changes in M to L cone ratios were primarily due to changes in cell fate specification, rather than dramatic loss or gain of M or L cone subtypes. As the "Late RA" condition did not affect M versus L cone subtype ratios (**Fig 3E and 3I**), but did increase cone density observed on day 200 (**S4F Fig**), addition of RA late in organoid development likely improves cone survival but does not affect fate specification. Addition of RA throughout development prevented normal development and observation of M or L cones at day 200, suggesting that continuous high RA is detrimental to overall cone or organoid health. Alternatively, as high RA early (**Fig 3E and 3H**) yields organoids enriched for M cones, late RA may specifically impair M cone survival.

Based on *M*- and *L-opsin* expression, cones were identified near the surface of all organoids in the presumptive ONL (**Fig 3F–3I**). During dissection and sectioning, the irregular shapes of organoids (**S3G–S3H Fig**) can result in a mix of cross and tangential sections, likely leading to the dispersed patterns of cones observed in some sections.

Together, these data suggest that RA is sufficient to induce M cones and suppress L cones early in retinal organoid development, but not at later time points when cones are already specified.

## Natural variation in cone ratios is associated with sequence divergence in RA signaling regulators

In parallel, we evaluated natural variation in the ratios of M and L cones in a human population sample. M and L cone ratios vary widely among people with normal color vision [11,26–29]. We quantified cone ratios using flicker-photometric electroretinogram (FP-ERG), which

measures and analyzes spectral sensitivities to 35 degrees of eccentricity [26], in 738 males with normal color vision. We studied males because variation at the X-linked *L/M-opsin* gene locus is a primary site for mutations causing red-green color blindness, and an increase in variation could be revealed in hemizygous conditions [30–32]. We observed a range of L cone ratios (mean = 59.1%; SD = 16.9%; **Fig 4A**). The association between FP-ERG and % L is nonlinear and as 100% is approached, very small changes in measured spectral sensitivities are associated with relatively large changes in estimates of % L, leading to some values that exceeded 100% upon normalization (**Fig 4A**). This extensive variation in the L and M cone ratios is in sharp contrast with the modest variation observed in the S cone ratio (8% to 12%) [33–39].

To identify a potential genetic basis of this variation in L and M cone ratios, we used a targeted sequencing approach to test for associations between cone cell ratios and genetic variation in mechanistic regulators of cone specification. Our sequencing targeted 21 gene regions, including the *L/M-opsin* gene locus (*OPN1LW/MW*), the *S-opsin* gene (*OPN1SW*), three RA regulatory genes (*RARA, NR2F2/COUP-TFII, CYP26A1/C1*), and 16 other genes with putative photoreceptor specification roles (*DIO1, DIO2, DIO3, PIAS3, RXRG, RORB, NR2F1/ COUP-TFI, OTX1, NEUROD1, THRB, NRL, OTX2, ONECUT1, NR2E3, ONECUT2, SALL3*) (**S5 Fig**). We generated custom-designed oligos and used a solution-based capture method for target enrichment. We barcoded and pooled samples, which we sequenced with 125 bp paired-end reads. We mapped the reads, genotyped the samples, and evaluated associations with L cone ratios (**Fig 4D**). In this Manhattan plot (**Fig 4D**), the position of each variant along the y-axis represents the significance of the association with the variation in L cone ratio and the position along the x-axis represents the genomic position at each locus.

After alignment and genotyping, we scanned for associations between variants in these genes and L cone ratios (**Fig 4D**). We identified a significant association between SNP rs372754794 ($\hat{\beta} = -18.54$, *p*-value = $1.67 \times 10^{-5}$) and variation in the L cone ratio (**Fig 4B– 4D**). Individuals with the T/G genotype displayed a lower % L cone ratio than individuals with the T/T genotype (**Fig 4B and 4C**).

The SNP rs372754794, which occurs within the targeted sequence spanning *NR2F2/ COUP-TFII*, lies in an intron in a noncoding RNA (*NR2F2-AS1*) immediately upstream of *NR2F2* (**Fig 4E and 4F**). The minor (G) allele, which is associated with reduced L cone ratio, was nearly exclusive to African American and African subjects within our sample (MAF = 0.0359; **Fig 4G**) and exhibited similar patterns of frequency differentiation in external data from the 1000 Genomes Project [40]. The association with L cone ratio was robust even when restricting analysis to African American subjects ($\hat{\beta} = -14.45$, *p*-value = $1.44 \times 10^{-4}$) (**Fig 4H**). Within African American populations from the 1000 Genomes Project, rs372754794 exhibited only modest linkage disequilibrium (LD) with other nearby variants (**Fig 4F**), such that the associated haplotype is confined to the *NR2F2-AS1* transcript.

NR2F2 is a nuclear receptor that mediates RA signaling [41]. *NR2F2* and the upstream *NR2F2-AS1* are expressed in human retinas and retinal organoids during development (**S6A and S6B Fig**) [16]. The associated SNP lies in a region of high transcription factor binding identified as a putative enhancer (**S6C Fig**) that is predicted to interact and regulate the promoters of *NR2F2* and/or *NR2F2-AS1* (**S6D Fig**). Alternatively, as part of an lncRNA transcript, the SNP could affect *NR2F2-AS1*-mediated regulation of *NR2F2* or other genes. The observed association at NR2F2 is consistent with a role for RA signaling in M versus L cone subtype specification.

The second strongest associated SNP (rs36102671) approached but did not achieve statistical significance upon multiple testing correction (Bonferroni correction *p*-value threshold = $1.73 \times 10^{-5}$). This SNP lies within the first intron of the *RARA* gene, which encodes

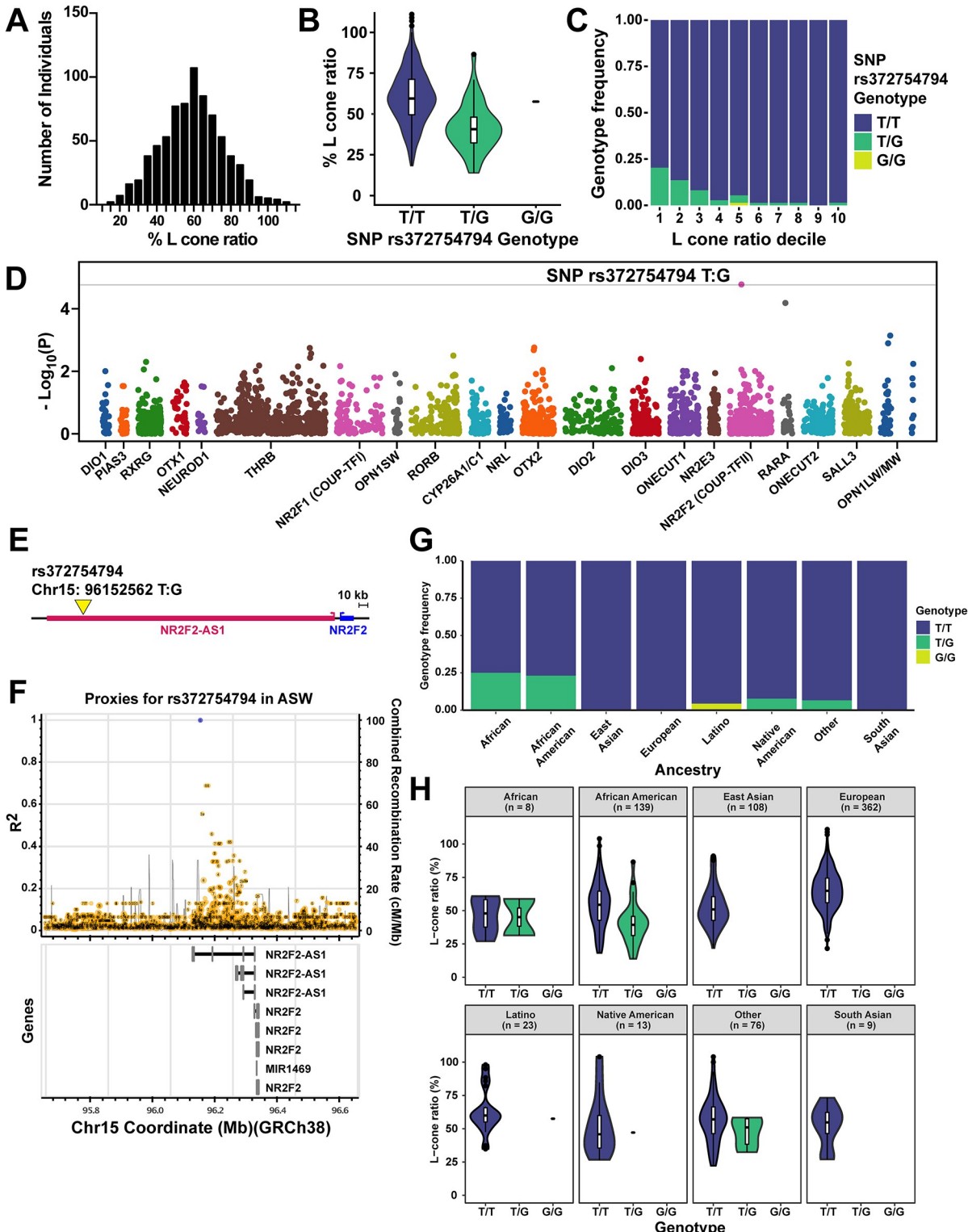

**Fig 4. Natural variation in cone ratios is associated with RA signaling regulation.** (**A**) Histogram of the ratios of % L from 738 human males with normal color vision. (**B**) L cone ratios for SNP rs372754794 genotypes. Original data sets are in S4 Data. (**C**) SNP rs372754794 genotype frequencies for L cone ratio deciles. Original data sets are in S4 Data. (**D**) Manhattan plot of the genetic variant *p*-values. Dots represent genetic variants in genes. Genetic variants above the gray line (Bonferroni corrected threshold) are significant ($p < 0.05$). The dots for each gene locus are presented in the same color (ex: *DIO1* = dark blue; *PIAS3* = orange). Along the X axis, the dots are spaced to scale as they occur in the genome. As *OPN1LW* and *OPN1MW* are nearby each other in the genome, the variants are represented together at the

*OPN1LW/MW* locus in the same dark blue color. Due to the highly similar sequences of *OPN1LW* and *OPN1MW*, many reads could not be mapped, resulting in the gap in the variants at the *OPN1LW/MW* locus. Original data sets are in S4 Data. **(E)** Location of SNP rs372754794 upstream at the *NR2F2* gene locus. *NR2F2* gene (blue); *NR2F2* antisense RNA (pink). Yellow arrow denotes the location of the SNP. **(F)** Local LD plot for rs37275494 based on data from an African American (ASW) population from the 1000 Genomes Project. Top indicates variants. Bottom indicates gene predictions. **(G)** Minor (G) allele ratio at rs37275494 by ancestry. Original data sets are in S4 Data. **(H)** Association of L:M ratio and rs372754794 stratified by self-reported ancestry. Original data sets are in S4 Data.

a receptor for RA (**Fig 4D**). The minor (A) allele, which segregates at low allele frequencies in globally diverse populations, is associated with decreased L cone ratios ($\hat{\beta}$ = −15.95, *p*-value = $6.54 \times 10^{-5}$) (**S7A Fig**). Within European populations, LD at this locus extends over an approximately 200 kb region of chromosome 17, including *RARA* and 7 other genes ($R^2 >$ 0.7). The SNP exhibits an association with altered splicing of RARA based on RNA sequencing of whole blood (**S7B Fig**) [42], suggesting that the haplotype modulates RARA function.

While validation and further dissection of these association results require investigation in a larger cohort, the association of RA-related candidates with changes in M and L cone ratios is consistent with a role for RA signaling in the specification of M and L cone subtypes in humans.

## Discussion

We found that *M-opsin*, but not *L-opsin*, is expressed in early developing fetal retinas. In adults, the earlier-born central and middle regions have higher proportions of M cones compared to the later-born periphery. RA-synthesizing *ALDHs* are highly expressed early in retinal development and decrease over time, suggesting that RA signaling is high during early development. RA is sufficient to promote M cone fate and suppress L cone fate in immature cones. Together, we propose that RA signaling is high early in human retinal development to promote the generation of M cones and low later to yield the generation of L cones (**Fig 5**).

Our observations suggest that RA signaling influences cone fate in immature cones. In human fetal retinas, *M-opsin* mRNA expression is readily observed on day 115, whereas expression of RA synthesis genes (i.e., *ALDHs*) is highest on day approximately 55 and then decreases. In human retinal organoids, addition of RA before day 60, well before opsin mRNA expression, induces M cones and suppresses L cones. RA signaling may act at the gene locus to regulate opsin expression directly or function upstream to influence M and L cone fates.

As (1) *M-opsin* or *L-opsin* were exclusively expressed in cones in fetal and adult retinas; and (2) RA promoted M cone fate early but did not convert L cones into M cones late, M cones and L cones likely arise directly from immature cones and do not undergo hybrid states. An

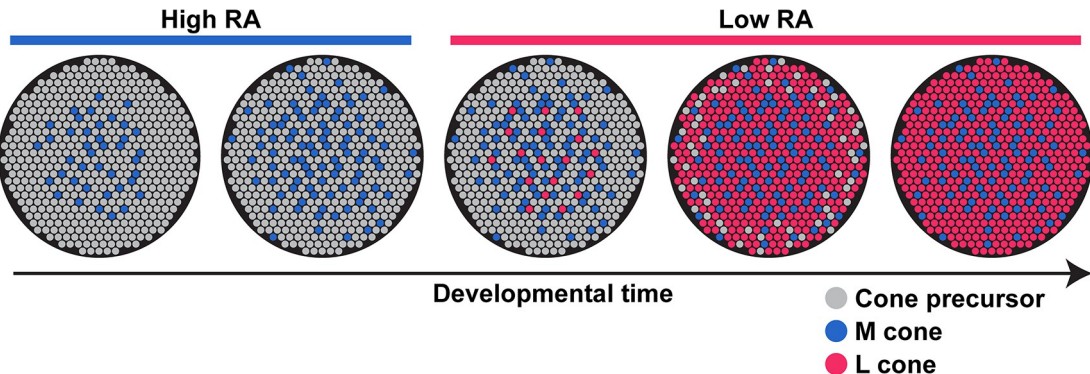

**Fig 5. Model—RA regulates the spatiotemporal patterning of M and L cones in the human retina.**

alternative is that all cones initially express *M-opsin* and then a subset switch and express *L-opsin*, perhaps undergoing a period of co-expression during late fetal development, a time point which was experimentally inaccessible. Though these details require further interrogation, our data clearly support temporal generation of M and L cones in the human retina.

*ALDH1A1* and *ALDH1A3* expression was observed in the central and peripheral retina on day 59 and decreased at later time points in both regions. *ALDH1A1* and *ALDH1A3* expression decreases to very low levels earlier in the central retina than periphery, reflecting the regionalized timing of retinal development, with the central region developing earlier than the periphery.

The arrangement of the LCR regulatory element and the *L*- and *M-opsin* genes in humans resembles an independently evolved gene array in zebrafish [20]. In zebrafish, long wavelength opsin expression is controlled by RA, which promotes expression of the proximal opsin gene late in development [21]. This contrasts with our findings that RA signaling promotes expression of the distal opsin early in humans. These observations suggest differences in the evolution of the regulation of RA signaling and the *cis*-regulatory logic controlling opsin expression, consistent with the independent origin of these opsin gene clusters.

RA appears to play multiple roles in human photoreceptor specification. In addition to its role in M versus L cone specification, RA regulates rod generation in human retinal organoids [43–45]. How RA coordinates the specification of cone and rod fate features is a key question for future studies.

Studying human development poses specific challenges including limited access to tissue, a lack of a controlled genetic background, and an inability to experimentally manipulate developing tissue. Vertebrate model organisms provided numerous insights into cell fate specification during retinal development. However, differences between these species and humans prevented interrogation of questions that can only be addressed in developing human tissue. Our studies show that human retinal organoids are a powerful model system to investigate developmental processes that are unique to humans, great apes, and Old World monkeys.

# Methods

## Cell lines

Cell lines used in this study are in **Table 1**.

**Table 1. Genotypes of cell lines.**

| Cell line | Sex | Source | Figures |
|---|---|---|---|
| HEK293 | Female | Kuruvilla lab, Johns Hopkins | **1C–1J, S1A, S1C** |
| WERI-Rb-1 | Female | ATCC | **S2B** |
| H7 ESC | Female | WA07, WiCell | **3E–3I, S4F–S4H** |
| H7 iCas9 ESC | Female | Zack lab, Johns Hopkins | **3E–3I, S4F–S4H** |

## HEK293 cell line maintenance

HEK293 cells were maintained in DMEM 4.5 g/L D-glucose, L-glutamine (11965084, Gibco) + 10% fetal bovine serum (16140071, Gibco) + 1× Penicillin-Streptomycin (30-002-CI, Corning) at 37˚C in a HERAcell 150i or 160i 5% $CO_2$ incubator (Thermo Fisher Scientific). Cells were split 1:4 to 1:10 every 2 to 3 days when approximately 80% confluent and kept in 6-well dishes or uncoated flasks using Trypsin (0.25%)-EDTA (0.02%) in HBSS (118-093-721, Quality Biological).

**Table 2. Opsin cDNA plasmids.**

| Plasmid | Source | Figures |
|---|---|---|
| OPN1MW NM_000513.2 | GenScript | 1C–1J, S1A, S1C |
| OPN1LW NM_020061.5 | GenScript | 1C–1J, S1A, S1C |

## Transfection of opsin cDNA into HEK293 cells

HEK293 cells were passaged 24 h prior to be <60% confluent on the day of transfection. Transfection was done with the Lipofectamine 3000 transfection kit (L3000001, Invitrogen). For a 10 cm dish, 7.5 μl lipofectamine 3000 was mixed with 125 μl Opti-MEM (31985062, Thermo Fisher Scientific) at room temperature (RT) for 5 min, and 5 μl of P3000, 0.5–1 μg DNA (Table 2), and 125 μl Opti-MEM was mixed at RT for 5 min as per manufacturer instruction. The 2 solutions were mixed and incubated at RT for 10 min. Media was aspirated from HEK293 cells and replaced with 8 mL of pre-warmed Opti-MEM. Lipofectamine mixture was added to cells and incubated at 37˚C in a HERAcell 150i or 160i 5% $CO_2$ incubator for 4 to 6 h. Transfection media was aspirated and replaced with HEK293 maintenance media (DMEM/10% FBS/1× Penn/Strep) and cells were allowed to recover overnight. Cells were adhered to slides coated with 10% Poly-L-lysine solution (0.1% w/v in $H_2O$) (P8920, Sigma) for 1 to 3 h at 37˚C in a HERAcell 150i or 160i 5% $CO_2$ incubator (Thermo Fisher Scientific). Cells were washed in 1× PBS and fixed in 10% neutral buffered formalin (HT501128, Sigma) for 30 min. Slides were either used immediately or ethanol dehydrated and stored at −80˚C.

## BaseScope RNA in situ hybridization

BaseScope RNA in situ hybridization was performed according to manufacturer's instructions (Advanced Cell Diagnostics) and modified with several changes. Adherent tissue culture cells or sections of human, fetal, or organoid tissue were allowed to come to RT from storage at −80˚C and rehydrated in 1× PBS. Samples were pretreated according to manufacturer's instructions: 10 min in RNAscope Hydrogen Peroxide followed by a wash in $dH_2O$ and then 2 washes in 1× PBS. The following step is an RNAscope Protease step which varies depending on tissue type. For HEK293 cells, RNAscope Protease III was applied at a 1:15 dilution in 1× PBS for 15 min in a humid chamber. For organoids and human eye samples, RNAscope Protease IV was applied for 20 min in a humid chamber. Samples were then washed in 1× PBS twice. Probes were added (Table 3) at the manufacturer suggested concentration to samples in the HybEZ Humidity Control rack with lid and inserted into the HybEZ oven for 2 h at 40˚C. Samples were washed twice in 1× RNAscope wash buffer for 2 min. Amplification and development washes were done with the manufacturer reagents at the recommended concentrations. All washes were done in the HybEZ Humidity Control rack with lid and insert, either at RT or at 40˚C in the HybEZ oven at the temperatures and time lengths according to Table 4, with 2× 2 min washes in 1× RNAScope Buffer conducted between each reagent wash.

**Table 3. BaseScope in situ probes.**

| Reagent type | Reagent | Source | Identifier | Additional information |
|---|---|---|---|---|
| BaseScope probe | BA-Hs-OPN1MW-1zz-st1 | Advanced Cell Diagnostics | 714201 | bp 902–941 NM_000513.2 |
| BaseScope probe | BA-Hs-OPN1LW-1zz-st-C2- | Advanced Cell Diagnostics | 716081-C2 | bp 880–921 NM_020061.5 |

**Table 4. BaseScope assay reagent washes.**

| Reagent | Temperature (˚C) | Time (min) |
|---|---|---|
| AMP 1 | 40 | 30 |
| AMP 2 | 40 | 30 |
| AMP 3 | 40 | 15 |
| AMP 4 | 40 | 30 |
| AMP 5 | 40 | 30 |
| AMP 6 | 40 | 15 |
| AMP 7 | RT | 30 |
| AMP 8 | RT | 15 |
| RED-A/B | RT | 10 |
| AMP 9 | 40 | 15 |
| AMP 10 | 40 | 15 |
| AMP 11 | RT | 30 |
| AMP 12 | RT | 15 |
| GREEN-A/B | RT | 10 |

Slides were baked for a minimum of 30 min at 65˚C in the HybEZ oven or until dry. Samples were preserved with VectaMount (Vector Laboratories, H-5000) mounting media and a sealed 1.5 coverslip.

## Imaging HEK293 RNA in situ hybridization

HEK293 cells were imaged using an EVOS XL Core Cell imaging system on brightfield with a 20× objective (Thermo Fisher).

## M/L-opsin immunohistochemistry and BaseScope in situ hybridization on HEK293 cells and adult human retina sections

HEK293 cells transfected with *M*- and/or *L-opsin* cDNA were allowed to come to RT after storage in 100% EtOH at 4˚C. Adult sections were allowed to come to RT from storage at −80˚C and dehydrated in a stepwise ethanol dehydration from 50%, 70%, and 100% EtOH. Samples were pretreated according to ACD Biotechne manufacturer's instructions: 10 min of RNAscope Hydrogen Peroxide at RT, followed by a wash in dH2O and then 2 washes in 1× PBS. Slides were then acclimatized for 10 s in 99˚C dH$_2$0, and then transferred to 99˚C RNAscope 1× Target Retrieval Reagent for 15 min using the Oster Double Tiered Food Steamer Warmer 5712. Slides were then transferred into dH$_2$0 for 15 s, and then transferred to 0.1% TWEEN (Sigma P7949). A primary antibody targeting both OPN1MW and OPN1LW (anti-rabbit Millipore ab5405 #3823289) was diluted 1:200 in Co-detection Antibody Diluent and added to the slides. All slides were incubated overnight in a humidity-controlled tray at 4˚C. Primary antibodies were washed off with 2 washes of 0.1% TWEEN for 2 min, and then postfixed by submerging in 10% neutral buffered formalin for 30 min at RT. Fixative was removed with 4 washes of 0.1% TWEEN for 2 min each. For HEK293 cells, RNAscope Protease III was applied at a 1:15 dilution in 1× PBS for 15 min in a humid chamber. For retina sections, RNAscope Protease IV was applied for 20 min in a humid chamber. Samples were washed in 1× PBS twice. Slides were washed twice in RNAscope 1× wash buffer for 2 min each. Co-Detection Blocker was applied to the slides and incubated for 15 min at 40˚C. Slides were then washed twice in RNAscope wash buffer for 2 min each, and then washed in 0.1% TWEEN for 2 min. Donkey anti-rabbit 647 secondary antibody (Invitrogen Thermo Fisher Scientific A31573

#2181018) was diluted 1:400 in Co-detection Antibody Diluent and applied to the samples overnight in a humidity control tray at 4˚C. Slides were then washed twice in 0.1% TWEEN for 2 min each, and co-stained in Hoechst 33342 (Biotium 40046 #10H0212, diluted 1:2,000 in 1× PBS).

## Imaging of M/L-opsin IHC and ISH

Slides were mounted in 1× PBS and the IHC staining was imaged the next day using either a Zeiss LSM 980 inverted microscope with an Axiocam 506 color camera system and a C Apo 40×/1.2 W DICIII lens or an Olympus IX81 inverted microscope equipped with a Hamamatsu Orca 4.0 v3 monochrome camera.

## ISH BaseScope of M and L-opsin in human adult retina and HEK293 cells

Coverslips from previous IHC assay were removed by soaking for 10 min in 5× SSC (Invitrogen by Thermo Fisher Scientific AM9763). BaseScope probes were then applied and amplified using the aforementioned protocol ("BaseScope RNA in situ hybridization"). Nuclei were stained using Hoechst 33342 (Biotium 40046 #10H0212, diluted 1:2,000 in 1× PBS).

## Imaging of ISH BaseScope M- and L-opsin in human adult retina and HEK293 cells

ISH BaseScope colorimetric staining was then imaged the day after protocol completion in brightfield using either a Zeiss LSM 980 inverted microscope with an Axiocam 506 color camera system and a C Apo 40×/1.2 W DICIII lens, or an Olympus IX81 inverted microscope equipped with a Hamamatsu Orca 4.0 v3 monochrome camera. A Cambridge Research & Instrumentation, Inc. (CRi) Micro*Color tunable RGB filter was inserted in the light path to allow fast sequential acquisition of red, green, and blue images, which were merged to form the final RGB data set.

## Semi-automated quantification of M- and L-opsin in adult retina

Representative pictures of *M-* and *L-opsin* in situ hybridization colorimetric staining from central, middle, and peripheral regions of the adult retina were deconvoluted in HALO Image Analysis Platform 3.5.3577 (Indica Labs) [46]. To visualize the chromogenic stain, we used the Deconvolution v1.1.7 module of the HALO Image Analysis Platform (Indica Labs) to separate out the colorimetric stains of the RGB images. Specifically, average RGB values from 9 to 16 pixels of each chromogenic stain were input into the brightfield algorithm to generate deconvoluted images. The chromogenic RED-A/B signal (generated by alkaline phosphatase against the BA-Hs-OPN1LW-1zz-st-C2 probe) was assigned red pseudo color and GREEN-A/B signal (generated by horseradish peroxidase against the BA-Hs-OPN1MW-1zz-st1 probe) was assigned green pseudo color. If retinal pigmented epithelium was present, this was assigned orange pseudo color. The deconvoluted staining was then used to count the number of M- and L-opsin-expressing foci using the Object Colocalization FL v2.1.4 module.

## Human adult retina tissue

Donor samples were acquired from the National Disease Research Interchange (NDRI) between 12.5 and 15.5 h postmortem and were flash frozen on dry ice after enucleation and stored at −80˚C. A human eye was allowed to come to RT in 1× PBS.

The anterior portion of the eye (cornea, iris, and lens) was removed. The posterior pole of the eye was butterflied in a petri dish, with 4 to 5 cuts made from the anterior ciliary body to

midway to the posterior, such that the retina and other tissue laid flat on the dish. A 5 mm punch biopsy tool (RBP-50, Robbins Instruments) was placed over the macula based on the presence of macular pigment in combination with presence of an avascular region to remove the central retina. Punches were taken immediately adjacent for the middle retina and immediately adjacent for the periphery. A total of 1 central punch, 3 middle punches, and 3 peripheral punches were taken from each eye (S3A Fig). Retinal punches were fixed for 45 min in 10% neutral buffered formalin (HT501128, Sigma) and washed in 1× PBS. The subsections of retina were mounted in Tissue-Tek O.C.T. compound (4583, Sakura), placed on dry ice to freeze, and stored at −80˚C. The retinal regions were cryosectioned in 20 μm sections. Slides were air dried for 6 h to overnight followed by a postfixation step of 15 min in 10% neutral buffered formalin (HT501128, Sigma) and washed in 1× PBS after drying. Slides were dried and stored at −80˚C for no more than 3 months before use.

## Imaging human adult RNA in situ hybridization

Samples were imaged in brightfield using a Zeiss LSM 980 inverted microscope with an Axiocam 506 color camera system using a C Apo 40×/1.2 W DICIII lens or an EVOS XL Core Cell imaging system on brightfield with a 20× air objective (Thermo Fisher).

## Human fetal retina tissue

Day 122 fetal retina (sex and left/right eye information unknown) and day 130 fetal eye (sex and left/right eye information unknown) were enucleated and flash frozen on dry ice and stored at −80˚C. The fetal eye was allowed to come to RT in 1× PBS. The anterior portion of the eye (lens, cornea, and iris) was removed with scissors. The posterior portion of the eye was fixed overnight in 10% neutral buffered formalin (HT501128, Sigma) and washed in 1× PBS. The whole posterior eye was mounted in Tissue-Tek O.C.T. compound (4583, Sakura), placed on dry ice to freeze, and stored at −80˚C. The eye was cryosectioned in 20 μm sections starting from the anterior side to the posterior, such that each section contains the nasal/temporal and dorsal/ventral information at each position along the anterior/posterior axis. For the day 130 fetal eye, sections were collected at 100 μm intervals, moving in a temporal to nasal direction. Sections were postfixed for 15 min in 10% neutral buffered formalin (HT501128, Sigma) and washed in 1× PBS, then air dried overnight. Slides were stored at −80˚C and used within 6 months.

## Imaging human fetal RNA in situ hybridization

Samples were imaged as overlapping tile regions in brightfield using a Zeiss LSM 980 inverted microscope with an Axiocam 506 color camera system using a C Apo 20×/1.2 W DICIII lens or C Apo 40×/1.2 W DICIII lens. Maximum intensity projections of z-stacks (4 to 8 optical sections, 2 μm step size) were rendered within Zen Blue Image Processing software.

## Orientation of RNA in situ hybridization staining within human fetal retinal tissue

To orient the chromogenic staining in the context of the developing retinal tissue, we counterstained a subset of fetal retina slides in Hoechst 33342 (Biotium 40046 #10H0212, diluted 1:2,000 in 1× PBS) for 10 min following the drying step of the BaseScope RNA in situ hybridization protocol. Slides were mounted in 1× PBS and immediately imaged on an EVOS M5000 imaging system using a 405 fluorescent LED light cube and an RGB channel with a 20× air objective (Thermo Scientific Invitrogen).

To visualize the nuclear counterstain alongside the Basescope chromogenic stain, we used the Indica Labs–Deconvolution v1.1.7 module of the HALO Image Analysis Platform version 3.5.3577 (Indica Labs) to separate the chromogenic stains of the RGB images. Specifically, average RGB values from 9 to 16 pixels of each chromogenic stain were input into the brightfield algorithm to generate deconvoluted images. The chromogenic RED-A/B signal (generated by alkaline phosphatase against the BA-Hs-OPN1LW-1zz-st-C2 probe) was assigned red pseudo color, GREEN-A/B signal (generated by horseradish peroxidase against the BA-Hs-OPN1MW-1zz-st1 probe) was assigned green pseudo color. If retinal pigmented epithelium was present, this was assigned orange pseudo color. Deconvoluted chromogenic images were then merged with nuclear counterstains in ImageJ software.

## IHC of fetal human retina sections

Day 122 human fetal retina sections were stained using the above IHC staining protocol ("M/L-opsin IHC and ISH BaseScope on HEK293 cells and adult human retina sections") with the following adjustments: In addition to OPN1MW and OPN1LW, primary antibodies targeting Ki67 (anti-mouse Santa Cruz Biotechnology sc-23900 #A1520) were applied to the fetal human retina sections. The following secondaries were used: donkey anti-rabbit 647 (Invitrogen Thermo Fisher Scientific A31573 #2181018), donkey anti-mouse 555 secondary antibody (Invitrogen Thermo Fisher Scientific A31570 #204536), and donkey anti-chick 488 secondary antibody (703-545-155 #156558 Jackson ImmunoResearch Laboratories) was diluted 1:400 in Co-detection Antibody Diluent. Nuclei were counter stained using Hoescht 33342 (Biotium 40046 #10H0212, diluted 1:2,000 in 1× PBS). Samples were imaged with a laser scanning confocal Zeiss LSM 980 inverted microscope using a C Apo 40×/1.2 W DICIII lens.

## WERI-Rb-1 cell line maintenance

WERI-Rb-1 retinoblastoma cells were maintained in RPMI 1640 Medium (11875135, Gibco) + 10% fetal bovine serum (16140071, Gibco) + 1× Penicillin-Streptomycin (30-002-CI, Corning) at 37°C in a HERAcell 150i or 160i 5% $CO_2$ incubator (Thermo Fisher Scientific). Cells were passaged every 4 days at approximately $1 \times 10^5$ to $2 \times 10^6$ cells/mL in uncoated flasks by pelleting at 150 g for 5 min and resuspending in fresh media.

## Bulk RNA sequencing samples

Fetal retina bulk RNA sequencing data were analyzed from Hoshino and colleagues [16]. Adult retina sequencing data were analyzed from Pinelli and colleagues [17]. WERI-Rb-1 samples were grown in control media or T3-treated media (100 nM T3 (T6397, Sigma) in RPMI + supplement media) for 4 days ($N$ = 1). RNA from individual samples was extracted using the Zymo Direct-zol RNA Microprep Kit (R2062, Zymo Research) according to manufacturer's instructions. Libraries were prepared using the Illumina TruSeq stranded mRNA kit and sequenced on an Illumina NextSeq 500 with single 75 bp reads. EP1 iPSC-derived organoids were previously grown and analyzed for Eldred and colleagues at time points ranging from day 10 to day 250 of differentiation [15].

## Bulk RNA sequencing analysis

Quantification of opsin expression in human fetal retina samples from [16,17] using code deposited https://github.com/bbrener1/johnston_retina: Expression of opsin genes was directly quantified from FASTQ files using Kallisto 0.44.0. The kallisto index was generated from the Gencode V27 human transcriptome, with the default kmer setting of 31. Reads were quantified in single-

**Table 5. Transcript positions for pileup analyses.**

| Exon | Base | | Transcript position | | Exon position | Absolute position on chrX (GRCh38) | |
|---|---|---|---|---|---|---|---|
| | LW | MW | LW | MW | | LW | MW |
| 2 | C | T | 254 | 276 | 81 | 154150737 | 154187851 |
| | A | G | 360 | 382 | 187 | 154150843 | 154187957 |
| | A | G | 391 | 413 | 218 | 154150874 | 154187988 |
| | C | A | 407 | 429 | 234 | 154150890 | 154188004 |
| 4 | T | C | 749 | 771 | 110 | 154154684 | 154191798 |
| | G | A | 757 | 779 | 118 | 154154692 | 154191806 |
| | C | G | 758 | 780 | 119 | 154154693 | 154191807 |
| | T | C | 759 | 781 | 120 | 154154694 | 154191808 |
| | A | G | 766 | 788 | 127 | 154154701 | 154191815 |
| 5 | A | G | 880 | 902 | 75 | 154156369 | 154193483 |
| | T | C | 883 | 905 | 78 | 154156372 | 154193486 |
| | T | G | 885 | 907 | 80 | 154156374 | 154193488 |
| | G | A | 888 | 910 | 83 | 154156377 | 154193491 |
| | A | T | 890 | 912 | 85 | 154156379 | 154193493 |
| | G | T | 895 | 917 | 90 | 154156384 | 154193498 |
| | C | A | 901 | 931 | 104 | 154156398 | 154193512 |
| | A | G | 905 | 935 | 108 | 154156402 | 154193516 |
| | T | C | 940 | 970 | 143 | 154156437 | 154193551 |
| | G | C | 944 | 974 | 147 | 154156441 | 154193555 |
| | A | T | 978 | 1008 | 181 | 154156475 | 154193589 |

ended mode with a specified length of 75 and standard deviation of 10. Quantification of opsin expression in organoid data: Organoid data was previously quantified and described in [15]. Expression levels were quantified using Kallisto (version 0.34.1) with the following parameters: "-b 100 -l 200 -s 10 -t 20 –single". The Gencode release 28 comprehensive annotation was used as the reference transcriptome [47]. For the analysis of opsin pileups, to examine the reads mapped to each opsin directly, reads were aligned to the opsin transcripts using Bowtie 2.3.4.3, and aligned files were processed with samtools 1.7 to produce pileups around selected locations in the transcripts. Pileups are a text-based format that quantifies base calls of aligned reads to a reference genome. For the quantification of human samples, a Bowtie2 index was generated from sequences ENST00000595290.5 and ENST00000369951.8 using the default settings. Pileups were generated for transcript positions given in **Table 5**.

## Stem cell line maintenance

H7 ESC (WA07, WiCell) and episomal-derived EP1.1 iPSC (Zack lab, Hopkins) were used for retinal organoid differentiation. Pluripotency of EP1.1 cells was evaluated previously with antibodies for NANOG, OCT4, SOX2, and SSEA2 [48]. Stem cells were maintained in mTeSR1 (85857, Stem Cell Technologies) on 1% (v/v) Matrigel-GFR (354230, BD Biosciences) coated dishes and grown at 37°C in a HERAcell 150i or 160i 10% $CO_2$ and 5% $O_2$ incubator (Thermo Fisher Scientific). Cells were passaged every 4 to 5 days according to confluence as in [48]. Cells were passaged with Accutase (SCR005, Sigma) for 7 to 12 min to be dissociated to single cells. Cells in Accutase were added 1:2 to mTeSR1 plus 5 μm Blebbistatin (Bleb, B0560, Sigma), pelleted at 150 g for 5 min, and suspended in mTeSR1 plus Bleb and plated at 5,000 to 15,000 cells per well in a 6-well plate. Media was replaced with mTeSR1 48 h following passage and every 24 h until passaged again. To minimize cell stress, no antibiotics were used.

## Organoid media

E6 supplement: 970 µg/mL Insulin (11376497001, Roche), 535 µg/mL holo-transferrin (T0665, Sigma), 3.20 mg/mL L-ascorbic acid (A8960, Sigma), 0.7 µg/mL sodium selenite (S5261, Sigma).

BE6.2 media for early retinal differentiation: 2.5% E6 supplement (above), 2% B27 Supplement (50×) minus Vitamin A (12587010, Gibco), 1% Glutamax (35050061, Gibco), 1% NEAA (11140050, Gibco), 1 mM Sodium Pyruvate (11360070, Gibco), and 0.87 mg/mL NaCl in DMEM (11885084, Gibco).

LTR (Long-Term Retina) media: 25% F12 (11765062, Gibco) with 2% B27 Supplement (50×) (17504044, Gibco), 10% heat inactivated FBS (16140071, Gibco), 1 mM Sodium Pyruvate, 1% NEAA, 1% Glutamax, and 1 mM taurine (T-8691, Sigma) in DMEM (11885084, Gibco).

## Retinoic acid treatments

For organoids, 1.04 µm all-trans retinoic acid (ATRA; R2625; Sigma) was thawed fresh in LTR media every 2 days, protected from light, and never freeze-thawed.

## Organoid differentiation and maintenance

Organoids were differentiated from H7 ESCs, H7iCas9 ESCs, or EP1.1 iPSCs (**Table 6**) as described in Eldred and colleagues with minor variations (**S4D Fig**) [15]. On day 0, pluripotent and well-maintained stem cells with minimal to no spontaneous differentiation were used for organoid aggregation. To aggregate, cells were passaged in Accutase (SCR005, Sigma) at 37°C for 12 min to ensure complete dissociation. Cells were seeded in 50 µls of mTeSR1 (85857, Stem Cell Technologies) at 3,000 cells/well into 96-well ultra-low adhesion round bottom Lipidure-coated plates (51011610, NOF) or ultra-low attachment microplate (7007, Corning). Corning plates were used upon the discontinuation of Lipidure plates. Cells were placed in hypoxic conditions (10% $CO_2$ and 5% $O_2$) for 24 h to enhance survival. Cells naturally aggregated by gravity over 24 h. On day 1, cells were moved to normoxic conditions (5% $CO_2$). On days 1 to 3, 50 µls of BE6.2 media containing 3 µm Wnt inhibitor (IWR1e: 681669, EMD Millipore) and 1% (v/v) Matrigel (354230, BD Biosciences) were added to each well. On days 4 to 9, 100 µl of media were removed from each well, and 100 µl of media were added. On days 4 and 5, BE6.2 media containing 3 µm Wnt inhibitor and 1% Matrigel was added. On days 6 and 7, BE6.2 media containing 1% Matrigel was added. On days 8 and 9, BE6.2 media containing 1% Matrigel and 100 nM Smoothened agonist (SAG: 566660, EMD Millipore) was added. On day 10, aggregates were transferred to 15 ml tubes, rinsed 3× in 5 ml DMEM (11885084, Gibco), and resuspended in BE6.2 with 100 nM SAG in untreated 10 cm polystyrene Petri dishes in a total of 12 ml of media. From this point on, media was changed every other day. Aggregates were monitored and manually separated if stuck together or to the bottom of the plate. On days 13 to 16, LTR media with 100 nM SAG was added. On day 16, retinal vesicles were manually dissected using sharpened tungsten needles, with cuts made to

**Table 6. Organoid replicates and cell lines.**

| Organoid treatment | Cell line | Replicates |
|---|---|---|
| No RA | H7 ESC | $N = 6$ |
| RA to day 60 | H7 ESC | $N = 6$ |
| RA to day 130 | H7 ESC, H7iCas9 ESC | H7 ESC ($N = 1$), H7 iCas9 ESC ($N = 2$) |
| Late RA | H7iCas9 ESC | $N = 5$ |

maximize the number of vesicles per organoid. On average, 1 to 2 cuts were made per organoid. After dissection, cells were transferred into 15 ml tubes and washed 3× with 5 ml DMEM (11885084, Gibco). On days 16 to 20, cells were maintained in LTR and washed 2× with 5 ml DMEM (11885084, Gibco), to remove dead cells, before being transferred to new plates. To increase survival and differentiation, 1.04 μm ATRA (R2625; Sigma) was added to LTR medium from days 20 to 43. Additional time windows of 1.04 μm were added depending on experimental conditions; 10 μm gamma-secretase inhibitor (DAPT, 565770, EMD Millipore) was added to LTR from days 28 to 42. Organoids were grown at low density (10 to 20 per 10 cm dish) to reduce aggregation. Periodically, organoids were culled from the plate based on absence of clear laminal structure indicating proper retinal organoid growth.

## Organoid preparation and cryosectioning

Organoids were fixed for 45 min in 10% neutral buffered formalin (HT501128, Sigma) and washed in 1× PBS. Organoids were placed in 25% sucrose in 0.1 M phosphate buffer solution overnight, and then mounted in Tissue-Tek O.C.T. compound (4583, Sakura), placed on dry ice to freeze, and stored at −80˚C. Organoids were cryosectioned in 10 μm sections. Slides were air dried for 6 h to overnight, then postfixated for 15 min in 10% neutral buffered formalin (HT501128, Sigma) and washed in 1× PBS. Slides were dried and stored at −80˚C and used within 3 months.

## Imaging organoid RNA in situ hybridization

Samples were imaged in brightfield using a Zeiss LSM 980 inverted microscope with an Axiocam 506 color camera system using a C Apo 40×/1.2 W DICIII lens or an EVOS XL Core Cell imaging system on brightfield with a 20× air objective (Thermo Fisher).

## Association study subjects

All experiments involving human subjects were conformed to the principles expressed in the Declaration of Helsinki and were approved by the University of Washington Human Subjects Review Committee. Subjects were 738 males who had previously passed several standard tests for color vision deficiency, including Ishihara's 24-plate test, Richmond HRR 2002 edition, and the Neitz Test of Color Vision. Subjects identified themselves as being either of African or African American ancestry ($N = 147$), Asian ancestry ($N = 108$), Caucasian ancestry ($N = 362$), Latino ancestry ($N = 23$), Mixed ($N = 62$), Native American ancestry ($N = 13$), South Asian ancestry ($N = 9$), or Other/Not Specified ($N = 14$). Subjects were dropped from the study if adequate quantity/quality of DNA or sequencing depth was not achieved.

The protocol for this human subject study was approved by the IRB at the University of Washington (IRB #41457). Written consent was obtained for all procedures for all subjects.

## Flicker-photometric electroretinogram (FP-ERG)

The proportion of L cones, expressed as the percentage of L plus M cones that were L, were determined for each subject from a combination of genetic analysis and the FP-ERG as described in [26,29]. Briefly, the spectral sensitivity of the individual was measured using control and test lights of specific wavelengths to illuminate a small portion of the retina. These conditions were designed to eliminate rod and S-cone contributions. While the test lights were flickered on and off, electrodes recorded the ERG neuronal responses to the stimulus. The spectral sensitivity was determined by adjusting the intensity of the test light until the ERG signal produced exactly matched that produced by the fixed intensity reference light. A correction

factor was applied to the estimated proportion of L cones to account for the 1.5 times greater contribution to the FP-ERG signal for each M cone compared to each L cone [28]. The association between FP-ERG and % L is nonlinear and as 100% is approached, very small changes in measured spectral sensitivities are associated with relatively large changes in estimates of % L. Thus, near 100%, small experimental errors led to some values that exceeded 100% upon normalization. These data combined with genetic analysis of the L and M cone genes to determine the spectral sensitivity of each opsin allowed for the determination of the % L cone proportion.

## Association study DNA preparation, pull-down, and sequencing

For each of these gene regions, we isolated DNA by targeting sequence intervals delineated by the end coordinate of the neighboring upstream gene to the start coordinate of the neighboring downstream gene using an oligo-based pull-down method (S5 Fig). If the neighboring upstream or downstream genes were less than 100 kb from the gene of interest, we expanded the pulled down region to include at least 100 kb of upstream and downstream sequence (S5 Fig). DNA was isolated from buccal swabs or whole blood using PureGene DNA extraction kits as described in [49]. DNA was then diluted to 10 ng/μl in low ETDA (USB Corporation, #75793). DNA was sheared to an average length of 250 bps using the Bioruptor Pico (Diagenode). Samples were stored on ice for 10 to 15 min prior to sonication. Sonication was performed for 7 cycles of on for 15 s, off for 90 s. After 3 cycles, the samples were briefly centrifuged before continuing with sonication for the remaining 4 cycles. Library preparation and hybridization pull-downs were performed according to the SeqCap EZ HyperCap Workflow User's Guide, version 2.0 following all recommended instructions. Libraries were created using the KAPA Hyper Prep Kit (KAPA Biosystems, #KK8504), using the KAPA Dual-Indexed Adapter Set (15 mM, KAPA Biosystems/Roche, #08 278 555 702). Probes for the pull-down were designed according to the Roche Nimble Design software, targeting the gene regions listed in (S5 Fig). Samples were sequenced on a Nova-seq, high-output, with 125 bp paired-end reads.

## Association study analysis

Paired-end reads for 738 samples were aligned to the human genome (build hg38) using BWA mem (v0.7.15) with default parameters. Duplicate reads were marked using Picard (v2.9.2). Each region was subdivided into nonoverlapping 1 kb windows starting at the upstream boundary using bedtools (v2.29.2). Mapped reads were genotyped on a region-by-region basis using Freebayes-parallel (v1.2.0), the argument "—use-best-n-alleles 4", and, for the region on chromosome X, "-p 1". Variant calls were filtered for quality greater than 10 times the number of additional observations, at least 1 alternative allele read on each strand, and at least 2 reads supporting the alternate allele balanced up- and downstream using vcflib (v1.0.0). Duplicate variants were then removed. Variants were filtered with PLINK (v1.90b6.12) for a minor allele frequency of at least 3.5% and a per-variant genotyping rate of at least 90%. Variants were pruned using LDAK (v5.0) with an LD cutoff of $R^2 < 0.75$ within 100 kb windows. A kinship matrix was calculated from the pruned variant set using LDAK, ignoring weights and a power of $-1$. $P$-values were calculated in LDAK from the pruned variant set, using the first 5 genotype eigenvectors as covariates to control for potential population stratification. Significant variants were defined using an alpha of 0.05 after Bonferroni correction.

## Quantification and statistical analysis

**HEK293 in situ hybridization quantification.** Images were tile scanned and all individual cells inside the region were manually counted using the Adobe Photoshop count tool. Cells were delineated by presence of pink, blue, or purple signal with a boundary drawn around the

signal. Images were contrasted to determine brightfield boundaries as needed. Non-expressing cells were not counted.

**Human adult retina in situ hybridization quantification.** For adult retinas, all serially sectioned punches of human retinas were imaged and counted manually and identified by presence of pink, blue, or purple signal using the Adobe Photoshop count tool. Exclusive expression of *M-opsin* or *L-opsin* was observed in cones. Co-expression of *M-opsin* and *L-opsin* was never observed in human retinas (purple signal). Cells were delineated by presence of pink, blue, or purple signal with a boundary drawn around the signal. Images were contrasted to determine brightfield boundaries. Average ratios were compared using a one-way ANOVA with Tukey's multiple comparisons test. Graphs and statistical tests were done in GraphPad Prism version 9.3.1 for Mac, GraphPad Software, San Diego, California, United States of America; www.graphpad.com. All error bars represent standard error of the mean (SEM).

**Quantification of IHC and ISH BaseScope of M- and L-opsin stained HEK293 cells and adult fetal retina sections.** IHC and ISH BaseScope images were manually aligned in Adobe Photoshop using the Hoescht nuclei stains. Images of the ISH BaseScope stain were used to count cells positive for *M-opsin* mRNA, *L-opsin* mRNA or both using the count tool. By overlaying the IHC stain, we determined which of these cells are also co-stained for M and L-opsin protein. Graphs and statistical tests were done in GraphPad Prism.

**Human opsin transcript pileups.** Transcripts per million (TPM) values were then used to generate graphs in GraphPad Prism.

**RNA sequencing graphs.** All graphs and statistical tests were generated using GraphPad Prism.

**Organoid quantification.** All sectioned organoids were imaged and counted manually using the Adobe Photoshop count tool. Organoids that had fewer than 150 cones ($n \leq 150$) were removed from analysis. A one-way ANOVA with Dunnett's multiple comparisons test was used to determine significance. All error bars represent SEM.

## Supporting information

**S1 Fig. In situ hybridization strategy to visualize *M-opsin* and *L-opsin* mRNA. (A)** Identification of HEK293 cells using Hoechst (light gray) in the experiment in **Fig 1I and 1J**. *M-opsin* (blue) and *L-opsin* (pink). Blue arrow indicates a cell expressing *M-opsin* mRNA only. Pink arrows indicate cells expressing *L-opsin* mRNA only. Purple arrow indicates a cell expressing both *M-opsin* mRNA and *L-opsin* mRNA. Black arrow indicates an untransfected cell. Note that the Hoechst signal is reduced in cells that express *M-opsin* mRNA and/or *L-opsin* mRNA, likely due to the colorimetric signal blocking the fluorescent signal. **(B)** Identification of retinal layers using Hoechst (light gray) in the experiment in **Fig 1K and 1L**. *M-opsin* (blue) and *L-opsin* (pink). No cones co-expressed *M-opsin* mRNA and *L-opsin* mRNA. ONL, outer nuclear layer; OPL, outer plexiform layer; INL, inner nuclear layer. **(C)** Quantification of *M-opsin* mRNA-expressing, *L-opsin* mRNA-expressing, or *M+L-opsin* mRNA-expressing cells that express M/L-opsin protein for the experiments conducted in HEK293 cells in **Fig 1I and 1J** and adult human retina in **Fig 1K and 1L**.
(PDF)

**S2 Fig. *M-opsin* and *L-opsin* expression in fetal retinas and WERI-RB-1 cells. (A)** 20 μm section of a fetal day 122 retina with expression of M-/L-opsin protein (magenta), Ki67 (cyan), and Hoechst/nuclei (white). M-/L-opsin protein is observed in the ONL. Ki67 expression indicates proliferating cells. **(B)** *M-* and *L-opsin* expression in control- and T3-treated WERI-Rb-1 retinoblastoma cells ($N = 1$ experiment). The WERI-Rb-1 retinoblastoma cell line expresses

*M-* and *L-opsin* at low levels [50]. T3, the active form of thyroid hormone, induces *M-* and *L-opsin* expression in WERI-Rb-1 cells [15,51]. Values indicate total pileup count normalized to total read count. Each data point indicates 1 nucleotide difference in *M-* or *L-opsin*. Original data sets are in S5 Data. **(C–I)** Analysis of *M-* and *L-opsin* mRNA expression in 130-day-old human fetal retina. **(C)** Example of *M-* and *L-opsin* mRNA expression in 130-day-old human fetal retina section. Anterior = left, posterior = right. Dorsal/ventral orientation is unknown. Colored boxes indicate regions shown in **(D–G)** zoomed in regions from **(C)**. The central region in **(F)** expressed opsin mRNA. The more peripheral regions in **(D)**, **(E)**, and **(G)** show no expression of opsin mRNA. **(H)** Quantification of cones expressing *M-* and *L-opsin* mRNA from the temporal to nasal sides of the globe. **(I)** Identification of retinal layers using Hoechst (light gray) in the experiment in (**S2C–S2H Fig**). *M-opsin* (blue), *L-opsin* (pink), and retinal pigmented epithelial (RPE, dark brown) colorimetric signals were deconvoluted to generate pseudo fluorescent images. These were overlaid with Hoechst nuclear counterstains to visualize retinal layers. ONL, outer nuclear layer; OPL, outer plexiform layer; INL, inner nuclear layer. Original data sets are in S5 Data.
(PDF)

**S3 Fig. Quantification of cells expressing *M-opsin* or *L-opsin* mRNA in adult retinas and validation. (A)** Schematic of retina with regions isolated using a 5 mm biopsy punch (as in **Fig 2G** with additional detail). White circle = optic nerve. Red lines = blood vessels. Yellow circle = macular pigment. **(B–E)** Validation of quantification of % of cells expressing *M-opsin* or *L-opsin* with HALO semi-automated image analysis software. **(B)** Images from 20 μm sections were probed for *M-opsin* (blue) and *L-opsin* (pink) mRNA. Images as in **Fig 2H–2J**. **(C)** HALO software deconvoluted the colorimetric *M-opsin* or *L-opsin* mRNA signals to generate pseudo fluorescent images (from **Fig S3B**). **(D)** Manually counted average ratios of M and L cones as percent of M/L total cones across 3 individuals. One-way ANOVA with Tukey's multiple comparisons test: Center L versus Middle L = no significance; Center L versus Periphery L $p < 0.01$; Middle L versus Periphery L $p < 0.01$; ** indicates $p < 0.01$. Data as in **Fig 2K** for comparison to **Fig S3E**. **(E)** HALO semi-automated software analysis of single deconvoluted representative images from the center, middle, and periphery regions (**Fig S3C**), showed similar ratios of M and L cones as manually scored retinas (**Fig 2K**). **(F–H)** Ratios of M and L cones as percent of M/L total cones for each individual; $n > 850$ cones for each region for each individual. Averages ratios are shown in **Fig 2K**.
(PDF)

**S4 Fig. *CYP26A1/B1/C1* expression during fetal retinal development and organoid differentiation experiments.** (A–C) Expression of *CYP26A1*, *CYP26B1*, and *CYP26C1* in fetal human retinas by day of gestation and retinal region. CPM, log counts per million. Analyzed from [16]. Error bars for the 2 samples from fetal day 94 indicate SEM. Original data sets are in S3 Data. (A) Whole retina. (B) Central retina. (C) Periphery. (D) Protocol for human retinal organoid differentiation, adapted from [15]. (E) Expression of *THRB* (cone marker) and *NRL* (rod marker) during retinal organoid development. TPM, transcripts per million. Analyzed from [15]. Original data sets are in S6 Data. (F) No significant differences in overall densities of M + L cones at day 200 in early RA treatment conditions (as in Fig 3F–3H) (Dunnett's multiple comparison's test, against "No RA" control: "RA to day 60" $p = 0.98$, "RA to day 130" $p = 0.32$). Significant difference between "No RA" and "Late RA" conditions (Dunnett's multiple comparison's test, * indicates $p < 0.05$) (as in Fig 3I). Error bars indicate SEM. Individual circles represent individual organoids. Original data sets are in S3 Data. (G) Representative brightfield image of a retinal organoid in "RA to day 130" conditions. (H) Representative

brightfield image of a retinal organoid in "Late RA" conditions.
(PDF)

**S5 Fig. Genes tested in association study.** Gene names and corresponding chromosome coordinates for human genome assembly GRCh38.
(PDF)

**S6 Fig. Expression of *NR2F2* and *NR2F2-AS1* during development. The rs372754794 SNP at the *NR2F2/NR2F2-AS1* locus lies in a putative regulatory region. (A)** Expression of *NR2F2* and *NR2F2-AS1* in human fetal retinas, analyzed from [16]. Original data sets are in S3 Data. **(B)** Expression of *NR2F2* and *NR2F2-AS1* in human retinal organoids, analyzed from [15]. Original data sets are in S6 Data. **(C)** ReMap ChIP-seq database [52] shows that the rs372754794 SNP lies in an enhancer based on transcription factor binding. Each colored line indicates ChIP-seq binding data for a different transcriptional regulator. The ReMap density shows the density of the peaks overlap. **(D)** GeneHancer database [53] shows that the rs372754794 SNP neighbors a region predicted to physically interact and regulate *NR2F2* and/ or *NR2F2-AS1*.
(PDF)

**S7 Fig. The rs36102671 SNP at the *RARA* locus is associated with differences in cone ratio. (A)** Association of L:M ratio and rs36102671 stratified by self-reported ancestry. Original data sets are in S7 Data. **(B)** rs36102671 association with altered splicing of RARA in whole blood from the Genotype Tissue Expression Project (GTEx). Original data sets are in S7 Data.
(PDF)

**S1 Data. Data that underlies Fig 1J, 1K.**
(XLSX)

**S2 Data. Data that underlies Figs 2C–2F and 2K, and S3D–S3H.**
(XLSX)

**S3 Data. Data that underlies Figs 3A–3C and 3E, S4A–S4C, S4F, and S6A.**
(XLSX)

**S4 Data. Data that underlies Fig 4A–4D and 4G, and 4H.**
(XLSX)

**S5 Data. Data that underlies S2B and S2H Fig.**
(XLSX)

**S6 Data. Data that underlies S4E and S6B Figs.**
(XLSX)

**S7 Data. Data that underlies S7A and S7B Fig.**
(XLSX)

## Acknowledgments

We dedicate this work to the memory of James Taylor, a great scientist and an even better person.

## Author Contributions

**Conceptualization:** Sarah E. Hadyniak, Kiara C. Eldred, James Taylor, Robert J. Johnston, Jr.

**Data curation:** Boris Brenerman, Rajiv C. McCoy, Michael E. G. Sauria, James Taylor.

**Formal analysis:** Boris Brenerman.

**Funding acquisition:** Robert J. Johnston, Jr.

**Investigation:** Sarah E. Hadyniak, Joanna F. D. Hagen, Kiara C. Eldred, Katarzyna A. Hussey, Rajiv C. McCoy, James A. Kuchenbecker, Maureen Neitz.

**Methodology:** Sarah E. Hadyniak, Joanna F. D. Hagen, Kiara C. Eldred, Katarzyna A. Hussey, Ian Glass, Maureen Neitz.

**Project administration:** Robert J. Johnston, Jr.

**Supervision:** Thomas Reh, Jay Neitz, James Taylor, Robert J. Johnston, Jr.

**Visualization:** Robert J. Johnston, Jr.

**Writing – original draft:** Sarah E. Hadyniak, Robert J. Johnston, Jr.

**Writing – review & editing:** Sarah E. Hadyniak, Kiara C. Eldred, Robert J. Johnston, Jr.

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
