## [Editor Report · Decision Letter 0]

7 Feb 2023

Dear Dr Johnston, 

Thank you for submitting your manuscript entitled "Spatiotemporal specification of human green and red cones" for consideration as a Research Article by PLOS Biology.

Your manuscript has now been evaluated by the PLOS Biology editorial staff, as well as by an academic editor with relevant expertise, and I am writing to let you know that we would like to send your submission out for external peer review.

Once your full submission is complete, your paper will undergo a series of checks in preparation for peer review. After your manuscript has passed the checks it will be sent out for review. To provide the metadata for your submission, please Login to Editorial Manager (https://www.editorialmanager.com/pbiology) within two working days, i.e. by Feb 09 2023 11:59PM.

Kind regards,

Kris

Kris Dickson, Ph.D., (she/her)

Neurosciences Senior Editor/Section Manager

PLOS Biology

kdickson@plos.org

---

## [Decision Letter · Decision Letter 1]

13 Apr 2023

Dear Dr Johnston,

Thank you for your patience while your manuscript "Spatiotemporal specification of human green and red cones" was peer-reviewed at PLOS Biology. It has now been evaluated by the PLOS Biology editors, an Academic Editor with relevant expertise, and by several independent reviewers. 

In light of the reviews, which you will find at the end of this email, we would like to invite you to revise the work to thoroughly address the reviewers' reports.

As you will see below, the reviewers find the study potentially interesting however they have raised a number of important suggestions to strengthen and expand the conclusions of the study. We think these should be carefully addressed before we can consider your manuscript further for publication.

As a note, after discussion with the Academic Editor, we would not strictly require that you provide the data asked for in Reviewer 2's point 8 (the S opsin and rhodopsin expression analyses) - while interesting, we think this request may be beyond the scope of the current work.

As another editorial note, the Academic Editor has also provided some suggestions to improve the presentation of your data and to make the figures more easily accessible. I have appended these below my signature and we hope you find them helpful.

Given the extent of revision needed, we cannot make a decision about publication until we have seen the revised manuscript and your response to the reviewers' comments. Your revised manuscript is likely to be sent for further evaluation by all or a subset of the reviewers.

**IMPORTANT - SUBMITTING YOUR REVISION**

*Re-submission Checklist*

*Published Peer Review*

*PLOS Data Policy*

*Blot and Gel Data Policy*

Sincerely,

Lucas

Lucas Smith, Ph.D.

Associate Editor

PLOS Biology

lsmith@plos.org

COMMENTS FROM THE ACADEMIC EDITOR:

The data presentation in the figures can be improved to help the reader. For example, it might be worth settling on a single quantification style for graphs like Fig. 2B-E, J, 3C,I,K S1B, S2D - these might be more easily interpreted if presented as boxplots?

Graphs 3A, S2A,C should be made consistent, and include sensible error metrics;

Stylistically, it would also be helpful to use color throughout the figures in a more consistent fashion, with a focus on applying it where it can help understanding (ex M vs L). In some cases the color choices are a bit unhelpful - e.g. why are 3I,J green/orange?; why is 3A RGB?; why yellow in 3B, K doesn't need color at all, but if so, this particular color-code is a bit confusing - e.g. are we meant to perceptually group the blue-ish groups? I don't think this is the intention, but if it is, it might be better to plot them next to each other, seeing that the X axis is non-continuous anyway...

Fig S1 could also be tightened up a bit, in terms of space

REVIEWS:

Reviewer #1: This is an elegant study that reports the prevalence of M cones over L cones during early human development.

This was made possible thanks to the ingenious development of a highly sensitive probe that could differentiate between M and L-opsin mRNAs despite the high degree of similarity between them.

Using human stem cell derived retinal organoids, the paper also convincingly demonstrates that retinoic acid (RA) signalling is much more pronounced in early development and appears sufficient to explain the early prevalence of M cones and suppression of L cone fate.

Experiments are well described and presented.

I have only one small issue with Figure 2B, showing M-opsin mRNA changes over post conceptional days. There seems to be a decrease between Day 132 and Day 136. Is that significant? and if so, does it indicate that marks the onset of downregulation in the production of M opsin mRNA at that age?

The second part of the paper describes variations in L cone prevalence in a human population sample from various ethnic backgrounds, showing that African and African American backgrounds are associated with a reduction in the natural ratio of L cones. This was found to be associated with a noncoding polymorphism in the NR2F2 gene which is associated with RA signalling, once again demonstrating a link between RA signalling and the promotion of M cones.

I have no doubt about the validity of these findings. but that aspect of the paper is a little bit obscure to non-aficionados in the field of human genetics and would benefit from being presented in a slightly less jargonish style. The figures associated with that aspect of the study are really difficult to understand, with very little explanation in support.

Reviewer #2: In this study Hadyniak and colleagues have devised a method to estimate the relative mRNA expression of the medium (M-) and long (L-) wavelength sensitive cone photopigments (opsins) in human retina and retinal organoids. They use this tool to address an important question of when and how cones select between M - or L- subtype fates. Given the similarity in the protein sequence of M and L opsins, the authors designed unique nucleotide sequences that can specifically identify M- vs L-opsin. Using these probes the authors performed in situ hybridization to determine the expression pattern of M- and L-opsin in adult human retina and at different stages of fetal development. They validate this approach first in HEK cells transfected with M-opsin and L-opsin plasmid DNA followed by immunohistochemistry to confirm opsin protein expression. They next apply this method in post-mortem human retina and find that M-opsin expression precedes L-opsin expression with M-opsin mRNA starting to express ~ fetal day 115 and no L-opsin mRNA expression until midgestation. The authors also find that in adult human retina the ratio of L- to M-opsin expression varies across regions and attribute this to the temporal sequence of cone specification/development in central vs peripheral retina. To dissect the underlying mechanism of cone fate selection, the authors explore the role of retinoic acid (RA) pathway and find that not only is the expression of RA synthesizing enzymes high in early fetal retina but also that manipulating the duration of exposure to RA in the culture medium can alter the cone fate selection in human retinal organoids. Lastly, the authors address if the variability in the ratio of M and L cones previously reported in humans across population is related to RA signaling. Using a targeted sequencing approach the authors find a strong association of RA signaling related genes with differences in M and L cone ratios. Overall the study is interesting, the results are quite exciting, and paper is well-written. However, there are several concerns related to the experiments, results and figure presentation outlined below that need to be properly addressed before publication.

1. I am concerned about the quality of the human retinal tissue as well as the labeling pattern that is being used to analyze/count M and L opsin expressing cones. For eg. it is difficult to understand or delineate from the images of human retina where is the opsin expression localized. One can't distinguish where the photoreceptor soma, inner and outer segments are in Fig 1K, 2A, G-I. It would be nice to have the entire retinal cross-section including a bright field image. Can these ISH labelings be done in whole mount retina preparation instead of sections? This will allow for better and more precise counting and estimation of M and L cones along with a better understanding of the cone mosaic.

2. How do the authors explain the 2-fold decrease in M-opsin expression from fetal day 132 to 136 without any concomitant change in L-opsin expression? 

3. When estimating the M and L opsin expression in fetal retina especially >fetal day 110, which retinal location is being used to analyze? Previous studies have shown that at these fetal stages the fovea is starting to form and the central retina has a characteristic morphology with more cones packed in this region and expressing M/L opsin (Zhang et al Neuron 2020). Can the authors separately analyze central and peripheral retinal M vs L-opsin expression at these later timepoints?

4. It will be important to mention which retinal location (in eccentricity or visual angle) are being referred to as center, middle and periphery in Fig 2F-J & Suppl Fig 1C-E. From the images it is not clear except there is some difference in cone density between periphery and the other two locations. How far is middle from periphery? 

5. The authors find that peripheral human retina is largely dominated by L-opsin expression reaching as high as >90% in some cases. This is a striking observation and could potentially revise our understanding of colour processing in the peripheral human retina. In fact, the findings in this study is in contrast to previous functional measurements of cone mosaic in peripheral non-human primate retina (Field et al, Nature 2010) where the distribution of M and L cones is less skewed to one type. To validate the results, could the authors analyze multiple peripheral locations in both nasal and temporal retina? 

6. It will be important to demonstrate that the regional variation in the M- and L-opsin expression is also dependent on RA signaling by performing similar analysis as in Fig 3A, across different retinal locations.

7. I am a little puzzled about the overall expression pattern of M and L opsins in the retinal organoids. In figure panels 3D and E, the opsin expression is clumped or clustered in patches over the surface of the organoid, with large gaps in the middle even though in either case (panel D or E) the majority of the cones should express L or M opsin. Shouldn't the expression be more continuous across the organoid? This poses some basic questions such as what is the cone density in these organoids and also what is the ratio of rods to cones? The authors must use better pictures in general and overlay the images with bright field pictures so that it is clear what the readers are looking at. 

8. As a control can the authors show that S opsin and rhodopsin expression i.e. the fraction of rods or S cones, in the retinal organoids are not impacted during the RA treatment. 

9. In this study the authors have estimated the fraction of cells expressing M or L opsin from the fluorescence signal derived from the ISH. Does the fluorescence signal vary from cell to cell and across preparation? Can the authors plot a distribution of the raw fluorescence intensities across cones in the human retina? How do the authors determine the threshold of expression level or fluorescence signal for cell selection/counting? Details about this should be incorporated in the methods. 

Reviewer #3: The primary goal of the manuscript is to explore the mechanisms specifying M vs L cone subtype fate. The authors first designed specific probes to detect of the expression of highly similar M and L-opsin mRNA and examined the timing of M and L-opsin mRNA expression in human retina by in situ hybridization and by analysis of the existing RNA-seq datasets. The work suggests that M cones are generated before L cones. The study further adds that retinoid acid promotes M cone fate early in development to generate the pattern of M/L cones in human retina. Finally the authors evaluated natural variation in the ratios of M and L cones in human population and identified a SNP within NR2F2/COUP-TFII locus that is significantly associated with variation in the L cone ratio. They also noted a second but statistically non-significant association with a SNP in RARA gene. These findings are consistent with the role of RA signaling in specifying M and L cone subtypes in human retina. 

Overall, the study presents some interesting findings. However, the manuscript seems to be stitched together by combining different experiments, which are not fully explored. The data are limited and somewhat incomplete to justify the conclusions. The manuscript includes three main figures. Fig 1 primarily tests the specificity of probes. Fig 2 presents data from M- and L-opsin expression based on bioinformatic analysis of one published human fetal retina dataset and one small dataset of adult human retina. It must be mentioned that gene expression in humans shows extensive variations. It doesn't take long to check multiple other datasets that are published from fetal, organoid and adult retinas. In fact, published scRNA-seq data can also help resolve this issue. In situ shown is from only one fetal day 122 retina. Previous immunohistochemistry studies by Hendrickson and others showed expression of S-opsin even before L/M in the central foveal region but later only L/M opsin is detected in fovea, suggesting that a detailed analysis (and not a single timepoint) is required to make specific conclusions. Therefore, in situ probes don't add much to the information that is already obtained from computational work. Fig 3 focuses on RA signaling. The role of thyroid hormone and RA signaling is well documented by many studies. RA addition in organoids is interesting. But, several papers have shown the involvement of RA signaling in rod differentiation in retinal organoids. The data being shown is less than optimal to make a conclusion. The human genetics part is a valuable addition but lacks power and is shown in the supplement. The selection of SNPs also biases the association analysis. These studies are potentially very interesting but require a larger cohort and independent replication.

Additional comments:

1. The statement that "trichromacy is unique to primates among mammals" is incorrect. Some Marsupials are trichromats as well, see a few PMID: 24737644, 21151905, 16859843, 11967153 

2. Please provide another image for Fig .1 H, that correlates with panel C and I of Fig.1. 

3. In Figure 1K, a low magnification image of the retina section should be included as well. In Supplemental Figure 1A, DAPI staining should be performed to highlight the retinal layers. 

4. Fig.1I lacks negative control. In Fig.1 I-L How many cells were taken? Be specific with the number of cells counted. Provide a table in the supplementary about the accuracy/confidence of the method

5. Fig. 2A lacks controls. Another image with defined cell boundaries must be included. 

6. The authors should explain why they examined RA signaling. In relation to this question, in Figure 3C, what causes the high proportion of L cone in "No RA" condition? Also, the bar graph in Figure 3C does not show the purple "M+L" as indicated on the right.

7. Fig. 3G-D : include bright field image of retinal organoids. 

8. Is the effect of RA dose-dependent? Did the authors examine the effect of lower or higher concentrations of RA in organoids? 

9. The authors states that "addition of RA late in organoid development likely improves cone survival but not fate specification", but also that "organoids grown in supplemental RA throughout development failed to normally differentiate yielding minimal to no M or L cones". Can the authors explain the discrepancy? Did the organoids die in supplemental RA throughout development? Why did it repress differentiation?

10. The authors defined three main timeframe of cone development: immature cone generation, maturation, and terminal differentiation. Terminal differentiation usually includes morphogenesis, e.g., formation of outer segment, and the acquisition of phototransduction ability. How does it differ from "maturation", in terms of cellular and molecular changes?

11. Do human retinal organoids also display higher proportion of M cones in the center, as in human retina?

---

## [Decision Letter · Decision Letter 2]

6 Nov 2023

Dear Dr Johnston,

Thank you for your patience while we considered your revised manuscript "Spatiotemporal specification of human green and red cones" for publication as a Research Article at PLOS Biology. This revised version of your manuscript has been evaluated by the PLOS Biology editors, the Academic Editor and two of the original reviewers.

The reviewers are largely satisfied by the revision, however Reviewer 3 has a few minor points that we think should be addressed before publication. Based on the reviews we are likely to accept this manuscript, provided you satisfactorily address the remaining points raised by the reviewers. 

**IMPORTANT: As you address the reviewer requests, please also attend to the following editorial requests. 

1) TITLE: Reviewer 3 has suggested that you change the title. After some discussion within the team, we think the title might be improved by adding more specificity. Although we will leave it to you to decide how best to respond to this reviewer request, as an example (and if supported) , we might suggest the title be changed to something like ""Retinoic acid signaling drives spatiotemporal specification of human green and red cones"

2) FINANCIAL DISCLOSURES: In the relevant section of our online system, please update your financial disclosures statement to describe the role of any sponsors or funders in the study design, data collection and analysis, decision to publish, or preparation of the manuscript. If the funders had no role in any of the above, include this sentence at the end of your statement: "The funders had no role in study design, data collection and analysis, decision to publish, or preparation of the manuscript."

3) METHODS: Please move the methods section to the main text. 

4) BLURB: In the relevant section of our online system, please provide a blurb which (if accepted) will be included in our weekly and monthly Electronic Table of Contents, sent out to readers of PLOS Biology, and may be used to promote your article in social media. The blurb should be about 30-40 words long and is subject to editorial changes. It should, without exaggeration, entice people to read your manuscript. It should not be redundant with the title and should not contain acronyms or abbreviations.

5) ETHICS STATEMENT: Please update your methods section to include the specific Institutional Review Board (IRB) or an equivalent committee, that approved the use of human tissue in this study. Please also include any relevant approval numbers for your protocols. 

6) DATA NOT SHOWN: Please note that per journal policy, we do not allow the mention of "data not shown", "personal communication", "manuscript in preparation" or other references to data that is not publicly available or contained within this manuscript. >>Please either remove mention of these data or provide figures presenting the results and the data underlying the figure.

7) DATA: I see in your current data availability statement that bulk RNA sequencing data are available upon request and that these cannot be made publicly available due to consent requirements. While we are OK with you limiting the sharing of this raw data, we recommend that you deposit this restricted data to a repository that allows for controlled data access. Alternatively, if this is not possible, please provide a non-author institutional point of contact, such as a data access or ethics committee, helps guarantee long term stability and availability of data. Providing interested researchers with a durable point of contact ensures data will be accessible even if an author changes email addresses, institutions, or becomes unavailable to answer requests.

8) DATA: You may be aware of the PLOS Data Policy, which requires that all data be made available without restriction: http://journals.plos.org/plosbiology/s/data-availability. For more information, please also see this editorial: http://dx.doi.org/10.1371/journal.pbio.1001797

a. Supplementary files (e.g., excel). Please ensure that all data files are uploaded as 'Supporting Information' and are invariably referred to (in the manuscript, figure legends, and the Description field when uploading your files) using the following format verbatim: S1 Data, S2 Data, etc. Multiple panels of a single or even several figures can be included as multiple sheets in one excel file that is saved using exactly the following convention: S1_Data.xlsx (using an underscore).

b. Deposition in a publicly available repository. Please also provide the accession code or a reviewer link so that we may view your data before publication. 

>>Regardless of the method selected, please ensure that you provide the individual numerical values that underlie the summary data displayed in the following figure panels as they are essential for readers to assess your analysis and to reproduce it:

Fig 1J,K; Fig 2C-F,K; Fig 3A-D,E; Fig 4A-D,G-H; Fig S2B,H; Fig S3D-H; Fig 4A-C,E-F; Fig 6A-B; Fig S7A-B;

>>Please also ensure that figure legends in your manuscript include information on where the underlying data can be found, and ensure your supplemental data file/s has a legend.

>>Please ensure that your Data Statement in the submission system accurately describes where your data can be found.

We expect to receive your revised manuscript within two weeks. 

*Published Peer Review History*

*Press*

Sincerely,

Lucas

Lucas Smith, Ph.D.

Senior Editor,

lsmith@plos.org,

PLOS Biology

Reviewer remarks:

Reviewer #2: The authors have addressed most of my concerns with additional analysis, experiments and clarifications. The revised manuscript is sufficiently strengthened for publication. 

Reviewer #3: The manuscript is much improved after making suggested revisions. I am generally satisfied with the revised manuscript. However, some of the data including organoids and genetic association are suggestive. These are interesting observations that require further investigations. 

L/M fate choice and cone distribution in human retina are likely more complex with additional involvement of multiple signaling pathways and cell interactions. The authors should change the title of the manuscript and tone down a few sentences in the text to properly reflect the data being presented.

---

## [Editor Report · Decision Letter 3]

6 Dec 2023

Dear Bob,

Thank you for the submission of your revised Research Article "Retinoic acid signaling regulates spatiotemporal specification of human green and red cones" for publication in PLOS Biology and thank you for addressing the last reviewer and editorial requests in this revision. On behalf of my colleagues and the Academic Editor, Tom Baden, I am pleased to say that we can in principle accept your manuscript for publication, provided you address any remaining formatting and reporting issues. These will be detailed in an email you should receive within 2-3 business days from our colleagues in the journal operations team; no action is required from you until then. Please note that we will not be able to formally accept your manuscript and schedule it for publication until you have completed any requested changes.

**We do have a couple of last requests for you to attend to, along with any formatting requests to come: 

1 - I saw in your most recent cover letter that you are in the process of setting up a third party point of contact to facilitate access to some of the restricted raw data. We ask that you please update your data availability statement in our online system, with this information as soon as it is ready, as this will be required for publication. 

2 - As previously discussed, I replaced the previous version of your manuscript with the track changes version of which you provided to me over email. So you will need to update the manuscript file in our system, to include a 'clean' version, that is updated with all of the final changes. 

PRESS

Sincerely, 

Luke

Lucas Smith, Ph.D.

Senior Editor

PLOS Biology

lsmith@plos.org